# THE GEOMETRY OF INTEGRATION IN TEXT CLASSIFICATION RNNS

**Kyle Aitken,**[*,†]
Department of Physics
University of Washington
Seattle, WA

**Vinay V. Ramasesh,**[†]
Blueshift, Alphabet
Mountain View, CA

**Ankush Garg,**
Google Research
Mountain View, CA

**Yuan Cao,**
Google Research
Mountain View, CA

**David Sussillo,**
Google Research
Mountain View, CA

**Niru Maheswaranathan**
Google Research
Mountain View, CA

## ABSTRACT

Despite the widespread application of recurrent neural networks (RNNs), a unified understanding of how RNNs solve particular tasks remains elusive. In particular, it is unclear what dynamical patterns arise in trained RNNs, and how those patterns depend on the training dataset or task. This work addresses these questions in the context of text classification, building on earlier work studying the dynamics of binary sentiment-classification networks (Maheswaranathan et al., 2019). We study text-classification tasks beyond the binary case, exploring the dynamics of RNNs trained on both natural and synthetic datasets. These dynamics, which we find to be both interpretable and low-dimensional, share a common mechanism across architectures and datasets: specifically, these text-classification networks use low-dimensional attractor manifolds to accumulate evidence for each class as they process the text. The dimensionality and geometry of the attractor manifold are determined by the structure of the training dataset, with the dimensionality reflecting the number of scalar quantities the network remembers in order to classify. In categorical classification, for example, we show that this dimensionality is one less than the number of classes. Correlations in the dataset, such as those induced by ordering, can further reduce the dimensionality of the attractor manifold; we show how to predict this reduction using simple word-count statistics computed on the training dataset. To the degree that integration of evidence towards a decision is a common computational primitive, this work continues to lay the foundation for using dynamical systems techniques to study the inner workings of RNNs.

## 1 INTRODUCTION

Modern recurrent neural networks (RNNs) can achieve strong performance in natural language processing (NLP) tasks such as sentiment analysis, document classification, language modeling, and machine translation. However, the inner workings of these networks remain largely mysterious.

As RNNs are parameterized dynamical systems tuned to perform specific tasks, a natural way to understand them is to leverage tools from dynamical systems analysis. A challenge inherent to this approach is that the state space of modern RNN architectures—the number of units comprising the hidden state—is often high-dimensional, with layers routinely comprising hundreds of neurons. This dimensionality renders the application of standard representation techniques, such as phase portraits, difficult. Another difficulty arises from the fact that RNNs are monolithic systems trained end-to-end. Instead of modular components with clearly delineated responsibilities that can be understood and tested independently, neural networks could learn an intertwined blend of different mechanisms needed to solve a task, making understanding them that much harder.

---

[*]Work started while an intern at Google.
[†]Equal contribution.

Recent work has shown that modern RNN architectures trained on binary sentiment classification learn low-dimensional, interpretable dynamical systems (Maheswaranathan et al., 2019). These RNNs were found to implement an integration-like mechanism, moving their hidden states along a line of stable fixed points to keep track of accumulated positive and negative tokens. Later, Maheswaranathan & Sussillo (2020) showed that contextual processing mechanisms in these networks—e.g. for handling phrases like *not good*—build on top of the line-integration mechanism, employing an additional subspace which the network enters upon encountering a modifier word. The understanding achieved in those works suggests the potential of the dynamical systems perspective, but it remained to be seen whether this perspective could shed light on RNNs in more complicated settings.

In this work, we take steps towards understanding RNN dynamics in more complicated language tasks, illustrating recurrent network dynamics in multiple text-classification tasks with more than two categories. The tasks we study—document classification, review score prediction (from one to five stars), and emotion tagging—exemplify three distinct types of classification tasks. As in the binary sentiment case, we find integration of evidence to underlie the operations of these networks; however, in multi-class classification, the geometry and dimensionality of the integration manifold depend on the type of task and the structure of the training data. Understanding and precisely characterizing this dependence is the focus of the present work.

**Our contributions**

- We study three distinct types of text-classification tasks—*categorical*, *ordered*, and *multi-labeled*—and find empirically that the resulting hidden state trajectories lie largely in a low-dimensional subspace of the full state space.

- Within this low-dimensional subspace, we find a manifold of approximately stable fixed points[1] near the network trajectories, and by linearizing the network dynamics, we show that this manifold enables the networks to integrate evidence for each classification as they processes the sequence.

- We find $(N-1)$-dimensional simplex attractors[2] for $N$-class categorical classification, planar attractors for ordered classification, and attractors resembling hypercubes for multi-label classification, explaining these geometries in terms of the dataset statistics.

- We show that the dimensionality and geometry of the manifold reflects characteristics of the training dataset, and demonstrate that simple word-count statistics of the dataset can explain the observed geometries.

- We develop clean, simple synthetic datasets for each type of classification task. Networks trained on these synthetic datasets exhibit similar dynamics and manifold geometries to networks trained on corresponding natural datasets, furthering an understanding of the underlying mechanism.

**Related work**  Our work builds directly on previous analyses of binary sentiment classification by Maheswaranathan et al. (2019) and Maheswaranathan & Sussillo (2020). Apart from these works, the dynamical properties of *continuous-time* RNNs have been extensively studied (Vyas et al., 2020), largely for connections to neural computation in biological systems. Such analyses have recently begun to yield insights on discrete-time RNNs: for example, Schuessler et al. (2020) showed that training continuous-time RNNs on low-dimensional tasks led to low-dimensional updates to the networks' weight matrices; this observation held empirically in binary sentiment LSTMs as well. Similarly, by viewing the discrete-time GRU as a discretization of a continuous-time dynamical system, Jordan et al. (2019) demonstrated that the continuous-time analogue could express a wide variety of dynamical features, including essentially nonlinear features like limit cycles.

Understanding and interpreting learned neural networks is a rapidly-growing field. Specifically in the context of natural language processing, the body of work on interpretability of neural models is reviewed thoroughly in Belinkov & Glass (2018). Common methods of analysis include, for example, training auxiliary classifiers (e.g., part-of-speech) on RNN trajectories to probe the network's

---

[1]As will be discussed in more detail below, by fixed points we mean hidden state locations that are *approximately* fixed on time-scales of order of the average phrase length for the task at hand. Throughout this work we will use the term fixed point manifold to be synonymous with manifolds of slow points.

[2]A 1-simplex is a line segment, a 2-simplex a triangle, a 3-simplex a tetrahedron, etc. A simplex is *regular* if it has the highest degree of symmetry (e.g. an equilateral triangle is a regular 2-simplex).

representations; use of challenge sets to capture wider language phenomena than seen in natural corpora; and visualization of hidden unit activations as in Karpathy et al. (2015) and Radford et al. (2017).

## 2 SETUP

**Models** We study three common RNN architectures: LSTMs (Hochreiter & Schmidhuber, 1997), GRUs (Cho et al., 2014), and UGRNNs (Collins et al., 2016). We denote their $n$-dimensional hidden state and $d$-dimensional input at time $t$ as $\mathbf{h}_t$ and $\mathbf{x}_t$, respectively. The function that applies hidden state update for these networks will be denoted by $F$, so that $\mathbf{h}_t = F(\mathbf{h}_{t-1}, \mathbf{x}_t)$. The network's hidden state after the entire example is processed, $\mathbf{h}_T$, is fed through a linear layer to get $N$ output logits for each label: $\mathbf{y} = \mathbf{W}\mathbf{h}_T + \mathbf{b}$. We call the rows of $\mathbf{W}$ 'readout vectors' and denote the readout corresponding to the $i^{\text{th}}$ neuron by $\mathbf{r}_i$, for $i = 1, \dots, N$. Throughout the main text, we will present results for the GRU architecture. Qualitative features of results were found to be constant across all architectures; additional results for LSTMs and UGRNNs are given in Appendix E.

**Tasks** The classification tasks we study fall into three categories. In the ***categorical*** case, samples are classified into non-overlapping classes, for example "sports" or "politics". By contrast, in the ***ordered*** case, there is a natural ordering among labels: for example, predicting a numerical rating (say, out of five stars) accompanying a user's review. Like the categorical labels, ordered labels are exclusive. Some tasks, however, involve labels which may not be exclusive; an example of this ***multi-labeled*** case is tagging a document for the presence of one or more emotions. A detailed description of the natural and synthetic datasets used is provided in Appendices C and D, respectively.

**Linearization and eigenmodes** Part of our analysis relies on linearization to render the complex RNN dynamics tractable. This linearization is possible because, as we will see, the RNN states visited during training and inference lie near approximate fixed points $\mathbf{h}^*$ of the dynamics—points that the update equation leave (approximately) unchanged, i.e. for which $\mathbf{h}^* \approx F(\mathbf{h}^*, \mathbf{x})$.[3] Near these points, the dynamics of the displacement $\Delta\mathbf{h}_t := \mathbf{h}_t - \mathbf{h}^*$ from the fixed point $\mathbf{h}^*$ is well approximated by the linearization

$$\Delta\mathbf{h}_t \approx \mathbf{J}^{\text{rec}}\big|_{(\mathbf{h}^*, \mathbf{x}^*)} \Delta\mathbf{h}_{t-1} + \mathbf{J}^{\text{inp}}\big|_{(\mathbf{h}^*, \mathbf{x}^*)} (\mathbf{x}_t - \mathbf{x}^*), \tag{1}$$

where we have defined the recurrent and input Jacobians $J_{ij}^{\text{rec}}(\mathbf{h}, \mathbf{x}) := \frac{\partial F(\mathbf{h}, \mathbf{x})_i}{\partial h_j}$ and $J_{ij}^{\text{inp}}(\mathbf{h}, \mathbf{x}) := \frac{\partial F(\mathbf{h}, \mathbf{x})_i}{\partial x_j}$, respectively (see Appendix A for details).

In the linear approximation, the spectrum of $\mathbf{J}^{\text{rec}}$ plays a key role in the resulting dynamics. Each eigenmode of $\mathbf{J}^{\text{rec}}$ represents a displacement whose magnitude either grows or shrinks exponentially in time, with a timescale $\tau_a$ determined by the magnitude of the corresponding (complex) eigenvalue $\lambda_a$ via the relation $\tau_a := |\log|\lambda_a||^{-1}$. Thus, eigenvalues within the unit circle thus represent stable (decaying) modes, while those outside represent unstable (growing) modes. The Jacobians we find in practice almost exclusively have stable modes, most of which decay on very short timescales (a few tokens). Eigenmodes near the unit circle have long timescales, and therefore facilitate the network's storage of information.

**Latent semantic analysis** For a given text classification task, one can summarize the data by building a matrix of word or token counts for each class (analogous to a document-term matrix (Manning & Schutze, 1999), where the documents are classes). Here, the $i, j$ entry corresponds to the number of times the $i^{\text{th}}$ word in the vocabulary appears in examples belonging to the $j^{\text{th}}$ class. In effect, the column corresponding to a given word forms an "evidence vector", i.e. a large entry in an particular row suggests strong evidence for the corresponding class. Latent semantic analysis (LSA) (Deerwester et al., 1990) looks for structure in this matrix via a singular value decomposition (SVD); if the evidence vectors lie predominantly in a low-dimensional subspace, LSA will pick up on this structure. The top singular modes define a "semantic space": the left singular vectors correspond to the projections of each class label into this space, and the right singular vectors correspond to how individual tokens are represented in this space.

---

[3]Although the fixed point expression depends on the input $\mathbf{x}$, throughout this text we will only study fixed points with the zero input. That is, we focus on the autonomous dynamical system given by $\mathbf{h}_{t+1} = F(\mathbf{h}_t, \mathbf{0})$ (see Appendix A for details).

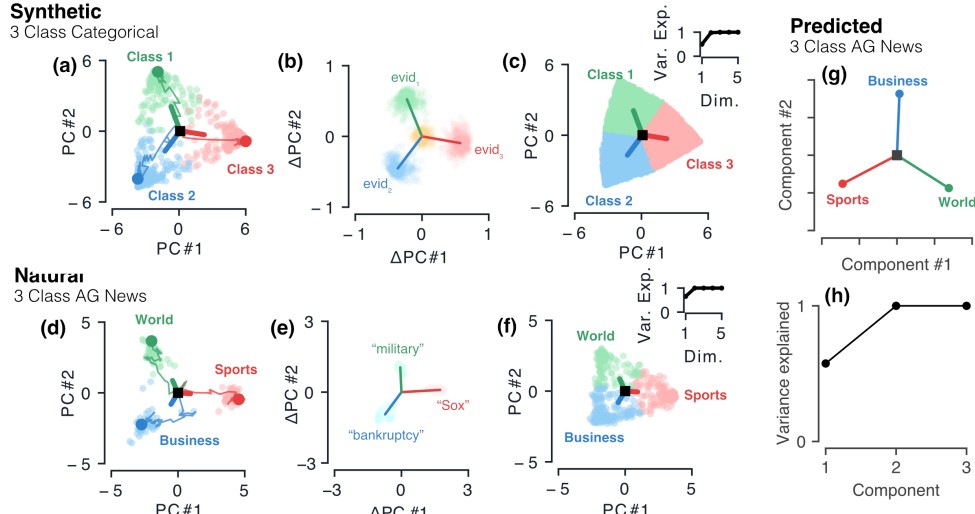

Figure 1: **Results from training GRUs on 3-class categorical data.** (a, d) Final hidden states, $\mathbf{h}_T$, of many test samples, colored by their label, with a few example hidden states trajectories shown explicitly. The initial state $\mathbf{h}_0$ is shown as a black square, and the three thick solid lines are the three readouts, colored by their respective class. (b, e) The hidden deflections, $\mathbf{h}_t - \mathbf{h}_{t-1}$ for $t = 1, \ldots, T$, from various words in the vocabulary, with the average deflection of each shown as a solid line. (c, f) Approximate fixed points, $\mathbf{h}^* \approx F(\mathbf{h}^*, \mathbf{x} = \mathbf{0})$, colored by their predicted label (see Appendix A.1 for details). The inset shows the variance explained as a function of number of PC dimensions. As in (a, d), the solid lines are readout vectors for each class. (g) The LSA score vectors projected into the top two variance dimensions. (h) Percentage of variance explained versus number of dimensions for LSA.

Below, we will show that RNNs trained on classification tasks pick up on the same structure in the dataset as LSA; the dimensionality and geometry of the semantic space predicts corresponding features of the RNNs.

**Regularization** While the main text focuses on the interaction between dataset statistics and resulting network dimensionality, regularization also plays a role in determining the dynamical structures. In particular, strongly regularizing the network can reduce the dimensionality of the resulting manifolds, while weakly regularizing can increase the dimensionality. Focusing on $\ell_2$-regularization, we document this effect for the synthetic and natural datasets in Appendicies D.1 and F, respectively.

## 3 RESULTS

### 3.1 CATEGORICAL CLASSIFICATION YIELDS SIMPLEX ATTRACTORS

We begin by analyzing networks trained on categorical classification datasets, with natural examples including news articles (AG News dataset) and encyclopedia entries (DBPedia Ontology dataset). We find dynamics in these networks which are largely low-dimensional and governed by integration. Contrary to our initial expectations, however, the dimensionality of the network's integration manifolds are *not* simply equal to the number of classes in the dataset. For example, rather than exploring a three-dimensional cube, RNNs trained on 3-class categorical tasks exhibit a largely *two*-dimensional state space which resembles an equilateral triangle (Fig. 1a, d). As we will see, this is an example of a pattern that generalizes to larger numbers of classes.

**Synthetic categorical data** To study how networks perform $N$-class categorical classification, we introduce a toy language whose vocabulary includes $N + 1$ words: a single evidence word "evid$_i$" for each label $i$, and a neutral word "neutral". Synthetic phrases, generated randomly, are labeled with the class for which they contain the most evidence words (see Appendix D for more details). This is analogous to a simple mechanism which classifies documents as, e.g., "sports" or "finance" based on whether they contain more instances of the word "football" or "dollar".

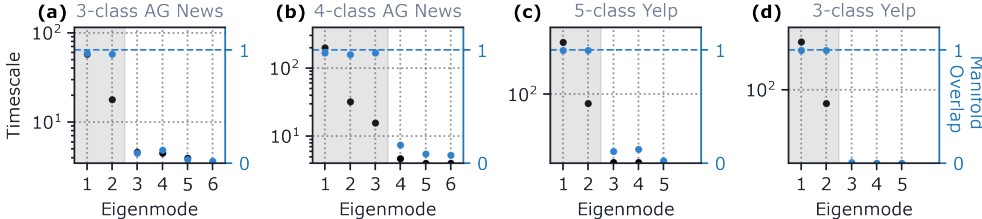

Figure 2: **Evidence integration is accomplished via eigenmodes with long timescales, aligned with the fixed-point manifold**. Each step along the x-axis represents a particular eigenmode of the dynamics, with corresponding eigenvalue $\lambda_a$; the black (left y-axis) points show the time constant (in units of tokens) associated with each mode, given by $\tau_a := |\log|\lambda_a||^{-1}$; the blue (right y-axis) points show the fraction of the mode's (right) eigenvector which lies in the fixed-point plane. From these plots, it is apparent that only a few modes (highlighted in gray) both: (i) do not decay appreciably on the time scale of the average document length, and (ii) aligned with, and create motion within, the fixed-point manifold. These are the integration modes responsible for accumulating evidence. We see **(a)** two integration modes in three-class categorical tasks, **(b)** three integration modes in four-class categorical tasks, and two integration modes in both **(c)** five-class ordered and **(d)** three-class ordered tasks. These plots characterize LSTMs, with other architectures shown in the Appendices.

The main features of the categorical networks' integration manifolds are clearly seen in the 3-class synthetic case. First, the dynamics are low-dimensional: performing PCA on the set of hidden states explored from hundreds of test phrases reveals that more than $97\%$ of its variance is contained in the top two dimensions. Projected onto these dimensions, the set of network trajectories takes the shape of an equilateral triangle (Fig. 1a). Diving deeper into the dynamics of the trained network, we examine the deflections, or change in hidden state, $\Delta\mathbf{h}_t$, induced by each word. The deflections due to evidence words "evid$_i$" align with the corresponding readout vector $\mathbf{r}_i$ at all times (Fig. 1b). Meanwhile, the deflection caused by the "neutral" word is much smaller, and on average, nearly zero. This suggests that the RNN dynamics approximate that of a two-dimensional integrator: as the network processes each example, evidence words move its hidden state within the triangle in a manner that is approximately constant across the phrase. The location of the hidden state within the triangle encodes the integrated, relative counts of evidence for each of the three classes. Since the readouts are of approximately equal magnitude and align with the triangle's vertices, the phrase is ultimately classified by whichever vertex is closest to the final hidden state. This corresponds to the evidence word contained the most in the given phrase.

**Natural categorical data**   Despite the simplicity of the synthetic categorical dataset, its working mechanism generalizes to networks trained on natural datasets. We focus here on the 3-class AG News dataset, with matching results for 4-class AG news and 3- and 4-class DBPedia Ontology in Appendix E. Hidden states of these networks, as in the synthetic case, fill out an approximate equilateral triangle whose vertices once again lie parallel to the readout vectors (Fig. 1d). While these results bear a strong resemblance to their synthetic counterparts, the manifolds for natural datasets are, unsurprisingly, less symmetric.

Though the vocabulary in natural corpora is much larger than the synthetic vocabulary, the network still learns the same underlying mechanism: by suitably arranging its input Jacobian and embedding vectors, it aligns an input word's deflection in the direction that changes relative class scores appropriately (Fig. 1e). Most words behave like the synthetic word "neutral", causing little movement within the plane; certain words, however, (like "football") cause a large shift toward a particular vertex (in this case, "Sports"). Again, the perturbation is relatively uniform across the plane, indicating that the order of words does not strongly influence the network's prediction.

In both synthetic and natural cases, the two-dimensional integration mechanism is enabled by a manifold of approximate fixed points, or *slow points*, near the network's hidden state trajectories, which allow the network to maintain its position in the absence of new evidence for all $t = 1, \ldots, T$. As the position within the plane encodes the network's integrated evidence, this maintenance is essential. In all 3-class categorical networks, we find a planar, approximately triangle-shaped manifold of fixed points which lie near the network trajectories (Fig. 1c, f); vertices of this manifold align with

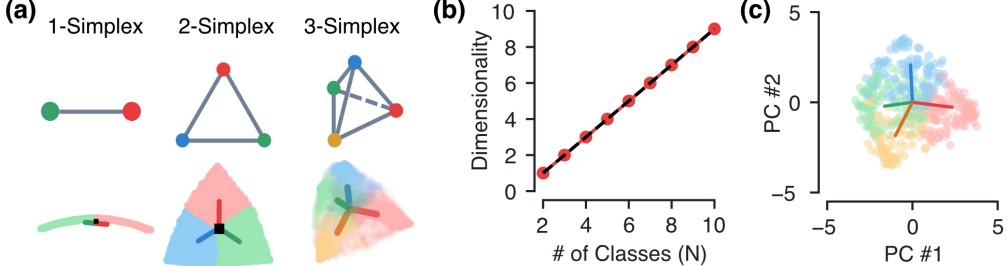

Figure 3: **Higher-dimensional simplexes in $N$-class categorical classification. (a)** Various simplex fixed-point manifolds seen in synthetic data for $N = 2, 3, 4$, colored by predicted label. **(b)** In red, the dimensionality of fixed-point manifolds from synthetic data as a function of number of classes, $N$. Here, dimensionality is the number of dimensions needed to capture more than $95\%$ of the variance in the fixed-point space. Each datapoint is an average over 10 initializations. The black dotted line is the dimensionality of an $(N-1)$-simplex. **(c)** The fixed point manifold and readouts for 4-class AG news, colored by predicted label.

the readout vectors. PCA reveals the dimensionality of this manifold to be very similar to that of the hidden states.

Since we find the network's hidden state trajectories always lie close to the fixed point manifold, we can use the fixed points' stability as an approximate measure of the network's ability to store integrated evidence. We check for stability directly by linearizing the dynamics around each fixed point and examining the spectra of the recurrent Jacobians. Almost all of the Jacobian's eigenvalues are well within the unit circle, corresponding to perturbations which decay on the timescale of a few tokens. Only two modes, which lie within the fixed-point plane, are capable of preserving information on timescales on the order of the mean document length (Fig. 2a). This linear stability analysis confirms our picture of a two-dimensional attractor manifold of fixed points; the network dynamics quickly suppress activity in dimensions outside of the fixed-point plane. Integration, i.e. motion within the fixed-point plane is enabled by two eigenmodes with long time constants (relative to the average phrase length).

**LSA predictions**   Intuitively, the two-dimensional structure in this three-class classification task reflects the fact that the network tracks relative score between the three classes to make its prediction. To see this two-dimensional structure quantitatively in the dataset statistics, we apply latent semantic analysis (LSA) to the dataset, finding a low-rank approximation to the evidence vectors of all the words in the vocabulary. This analysis (Fig. 1h) shows that two modes are necessary to capture the variance, just as we observed in the RNNs. Moreover, the class vectors projected into this space (Fig. 1g) match exactly the structure observed in the RNN readouts. The network appears to pick up on the same structure in the dataset's class counts identified by LSA.

**General $N$-class categorical networks**   The triangular structure seen in the 3-class networks above is an example of a general pattern: $N$-class categorical classification tasks result in an $(N-1)$-dimensional simplex attractor (Fig. 3a). We verify this with synthetic data consisting of up to 10 classes, analyzing the subspace of $\mathbb{R}^n$ explored by the resulting networks. More than $95\%$ of the variance of the hidden states is contained in $N-1$ dimensions, with the subspace taking on the approximate shape of a regular $(N-1)$-simplex centered about the origin (Fig. 1b). The readout vectors, which lie almost entirely within this $(N-1)$-dimensional subspace, align with the simplex's vertices. Mirroring the 3-class case, dynamics occur near a manifold of fixed points which also shaped like an $(N-1)$-simplex. This simplex geometry reflects the fact that to classify between $N$ classes, the network must track $N-1$ scalars: the relative scores for each class.

As a natural example of this simplex, the full, 4-class AG News dataset results in networks whose trajectories explore an approximate 3-simplex, or tetrahedron. The fixed points also form a 3-dimensional tetrahedral attractor (Fig. 3c). Additional results for 4-class natural datasets, which also yield tetrahedron attractors, are shown in Appendix E.

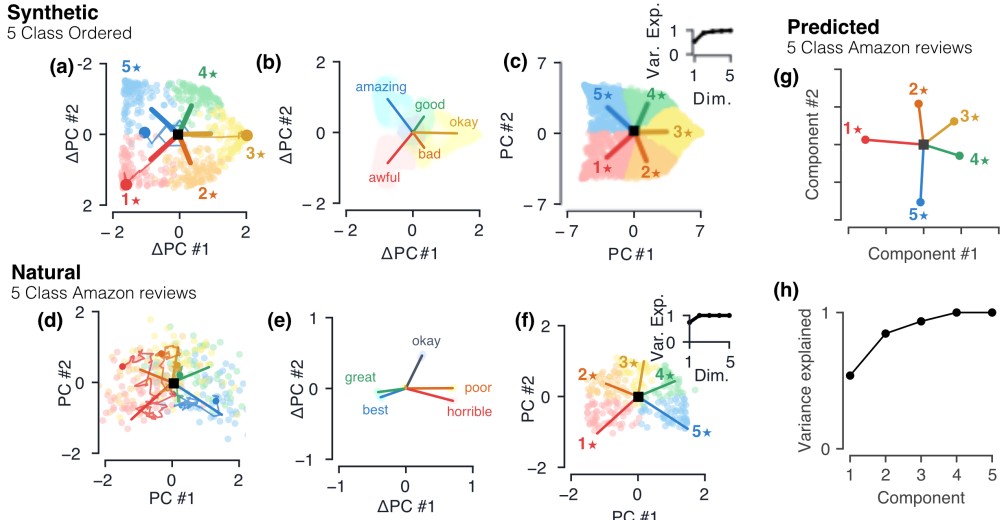

**Figure 4: Results from training GRUs on 5-class ordered data. (a, d)** Final hidden states, $\mathbf{h}_T$, of many test samples, colored by their label, with a few example hidden states trajectories shown explicitly. The initial state $\mathbf{h}_0$ is shown as a black square, and the three thick solid lines are the five readouts, colored by their respective class. **(b, e)** The hidden deflections, $\mathbf{h}_t - \mathbf{h}_{t-1}$ for $t = 1, \ldots, T$, from the various words in the vocabulary, with the average deflection of each shown as a solid line. **(c, f)** Approximate fixed points, $\mathbf{h}^* \approx F(\mathbf{h}^*, \mathbf{x} = \mathbf{0})$, colored by their predicted label (see Appendix A.1 for details). The inset shows the variance explained as a function of number of PC dimensions. As in (a, d), the solid lines are readout vectors for each class. **(g)** The LSA score vectors projected into the top two variance dimensions. **(h)** Percentage of variance explained versus number of dimensions for LSA.

## 3.2 Ordered classification yields plane attractors

Having seen networks employ simplex attractors to integrate evidence in categorical classification, we turn to ordered datasets, with Yelp and Amazon review star prediction as natural examples. Star prediction is a more finely-grained version of binary sentiment prediction that RNNs solve by integrating valence along a one-dimensional line attractor (Maheswaranathan et al., 2019). This turns out not to be the case for either 3-class or 5-class star prediction.

For a network trained on the 5-class Yelp dataset, we plot the two-dimensional projection of RNN trajectories while processing a test batch of reviews as well as the readout vectors for each class (Fig. 4d). Similar results for 3-class Yelp and Amazon networks are in Appendix E. The top two dimensions capture more than 98% of the variance in the explored states: as with categorical classification tasks, the dynamics here are largely low-dimensional. A manifold of fixed points that is also planar exists nearby (Fig. 4f). The label predicted by the network is determined almost entirely by the position within the plane. Additionally, eigenmodes of the linearized dynamics around these fixed points show two slow modes with timescales comparable to document length, separated by a clear gap from the other, much faster, modes (Fig. 2c, d). These two integration modes lie almost entirely within the fixed-point plane, while the others are nearly orthogonal to it.

These facts suggest that — in contrast to binary sentiment analysis — 5-class (and 3-class) ordered networks are *two-dimensional*, tracking two scalars associated with each token rather than simply a single sentiment score. As an initial clue to understanding what these two dimensions represent, we examine the deflections in the plane caused by particular words (Fig. 4f). These deflections span two dimensions—in contrast to a one-dimensional integrator, the word 'horrible' has a different effect than multiple instances of a weaker word like "poor." These two dimensions of the deflection vector seem to roughly correspond to a word's "sentiment" (e.g. good vs. bad) and "intensity" (strong vs. neutral). In this two-dimensional integration, a word like 'okay' is treated by the network as evidence of a neutral (e.g., 3-star) review.

Inspired by these observations, we build a synthetic ordered dataset with a word bank {amazing, good, okay, bad, awful, neutral}, in which each word now has a separate sentiment and

intensity score.[4] Labels are assigned to phrases based on both its total sentiment and intensity; e.g., phrases with low intensity and sentiment scores are classified as "3 stars", while phrases with high positive sentiment and high intensity are "5 stars" (see Appendix D.2 for full details). Networks trained on this dataset correspond very well to the networks trained on Amazon and Yelp datasets (Fig. 4a-c). Dynamics are largely two-dimensional, with readout vectors fanning out in the plane from five stars to one. Deflections from individual words correspond roughly to the sentiment and intensity scores, and the underlying fixed-point manifold is two-dimensional.

More generally, the appearance of a plane attractor in both 3-class and 5-class ordered classification shows that in integration models, relationships (such as order) between classes can change the dimensionality of the network's integration manifold. These relationships cause the LSA evidence vectors for each word to lie in a low-dimensional space. As in the previous section, we can see this low-dimensional in the dataset statistics itself using LSA, showing that two singular values explain more than 95% of the variance (Fig. 4f). Thus, the planar structure of these networks, with dimensions tracking both (roughly) sentiment and intensity, is a consequence of correlations present in the dataset itself.

## 3.3   MULTI-LABELED CLASSIFICATION YIELDS INDEPENDENT ATTRACTORS

So far, we have studied classification datasets where there is only a single label per example. This only requires networks to keep track of the *relative* evidence for each class, as the overall evidence does not affect the classification. Put another way, the softmax activation used in the final layer will normalize out the total evidence accumulated for a given example. This results in networks that, for an $N$-way classification task, need to integrate (or remember) at most $N-1$ quantities as we have seen above. However, this is not true in multi-label classification. Here, individual class labels are assigned independently to each example (the task involves $N$ independent binary decisions). Networks trained on this task *do* need to keep track of the overall evidence level.

To study how this changes the geometry of integration, we trained RNNs on a multi-label classification dataset, GoEmotions (Demszky et al., 2020). Here, the labels are emotions and a particular text may be labeled with multiple emotions. We trained networks on two reduced variants of the full dataset, only keeping two or three labels. The results for three labels are detailed in Appendix E.5. For the two-class version, we only kept the labels "admiration" and "approval", and additionally resampled the dataset so each of the $2^2 = 4$ possible label combinations were equally likely. We found that RNNs learned a two-dimensional integration manifold where the readout vectors span a two-dimensional subspace (Fig. 5d), rather than a one-dimensional line as in binary classification. Across the fixed point manifold, there were consistently two slow eigenvalues (Fig. 5e), corresponding to the two integration modes. Similar to the previous datasets, increasing $\ell_2$ regularization would (eventually) compress the dimensionality, again measured using the participation ratio (Fig. 5f). Notably, GoEmotions is a highly-imbalanced dataset; we found that balancing the number of examples per class was important to observe a match between the synthetic and the natural dynamics.

The synthetic version of this task classifies a phrase as if it were composed of two independent sentiment analyses (detailed in Appendix D.3). This is meant to represent the presence/absence of a given emotion in a given phrase, but ignores the possibility of correlations between certain emotions. After training a network on this data, we find a low-dimensional hidden-state and fixed-point space that both take on the shape of a square (Fig. 5a, c). The deflections of words affecting independent labels act along orthogonal directions (Fig. 5b).

These results suggest that integration manifolds are also found in RNNs trained on multi-labeled classification datasets. Moreover, the geometry of the corresponding fixed points and readouts is different from the exclusive case; instead of an $(N-1)$-dimensional simplex we get an $N$-dimensional hypercube. Again, this makes intuitive sense given that the networks must keep track of $N$ independent quantities in order to solve these tasks.

---

[4]Interestingly, a synthetic model that only tracks sentiment fails to match the dynamics of natural ordered data for $N > 2$. We take this as further evidence that natural ordered datasets classify based on *two*-dimensional integration. This simple model still produces surprisingly rich dynamics that we detail in Appendix D.2.

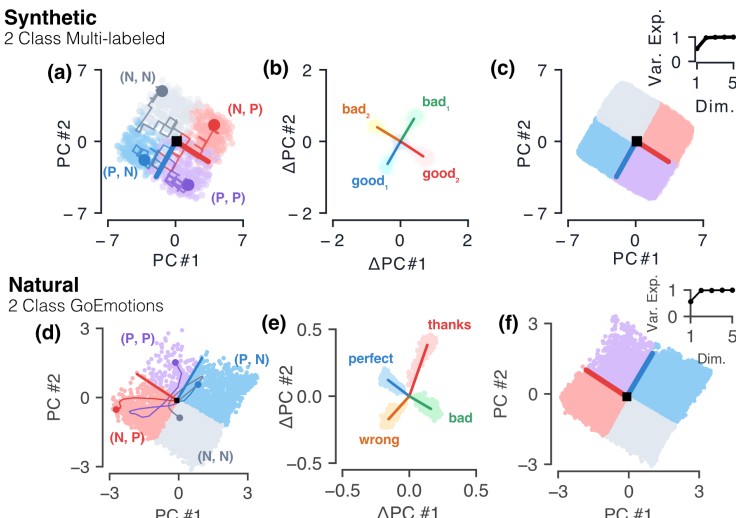

Figure 5: **GRUs trained on multi-labeled synthetic (a-c) and natural (d-f) GoEmotions dataset with only two labels.** (a, d) Final hidden states, $\mathbf{h}_T$, of a synthetic network, colored by their label, with a few example trajectories. The two thick solid lines are readout vectors. The network classifies examples using a 2D plane attractor. (b, e) The hidden deflections, $\mathbf{h}_t - \mathbf{h}_{t-1}$ for $t = 1, \ldots, T$, from the various words in the vocabulary, with the average deflection of each shown as a solid line. (c, f) Approximate fixed points, $\mathbf{h}^* \approx F(\mathbf{h}^*, \mathbf{x} = \mathbf{0})$, colored by their predicted label; these form a square (cf. the triangle in Fig. 1). The inset shows the variance explained (two dimensions are sufficient to capture nearly all of the variance).

## 4 DISCUSSION

In this work we have studied text classification RNNs using dynamical systems analysis. We found integration via attractor manifolds to underlie these tasks, and showed how the dimension and geometry of the manifolds were determined by statistics of the training dataset. As specific examples, we see $(N-1)$-dimensional simplexes in $N$-class categorical classification where the network needs to track relative class scores; 2-dimensional attractors in ordered classification, reflecting the need to track sentiment and intensity; and $N$-dimensional hypercubes in $N$-class multi-label classification.

We hope this line of analysis — using dynamical systems tools to understand RNNs — builds toward a deeper understanding of how neural language models perform more involved tasks in NLP, including language modeling or translation. These tasks cannot be solved by a pure integration mechanism, but it is plausible that integration serves as a useful computational primitive in RNNs more generally, similar to how line attractor dynamics serve as a computational primitive on top of which contextual processing occurs (Maheswaranathan & Sussillo, 2020).

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

## A  METHODS

### A.1  FIXED-POINTS AND LINEARIZATION

We study several RNN architectures and we will generically denote their $n$-dimensional hidden state and $d$-dimensional input at time $t$ as $\mathbf{h}_t$ and $\mathbf{x}_t$, respectively. The function that applies hidden state update for these networks will be denoted by $F$, so that $\mathbf{h}_t = F(\mathbf{h}_{t-1}, \mathbf{x}_t)$. The $N$ output logits are a readout of the final hidden state, $\mathbf{y} = \mathbf{W}\mathbf{h}_T + \mathbf{b}$. We will denote the readout corresponding to the $i$th neuron by $\mathbf{r}_i$, for $i = 1, \dots, N$.

We define a fixed point of the hidden-state space to satisfy the expression $\mathbf{h}^* = F(\mathbf{h}^*, \mathbf{x})$. This definitions of fixed-points is inherently $\mathbf{x}$-dependent. In this text, we focus on fixed points of the network in for zero input, i.e. when $\mathbf{x} = \mathbf{0}$. We will also be interested in finding points in hidden state space that only satisfy this fixed point relation approximately, i.e. $\mathbf{h}^* \approx F(\mathbf{h}^*, \mathbf{x})$. The slowness of the approximate fixed points can be characterized by defining a loss function $q := \frac{1}{n} \|\mathbf{h} - F(\mathbf{h}, \mathbf{x})\|_2^2$. Throughout this text we use the term *fixed point* to include these approximate fixed points as well.

Expanding around a given hidden state and input, $(\mathbf{h}^e, \mathbf{x}^e)$, the first-order approximation of $F$ is

$$\mathbf{h}_t \approx F(\mathbf{h}^e, \mathbf{x}^e) + \mathbf{J}^{\text{rec}}\big|_{(\mathbf{h}^e, \mathbf{x}^e)} (\mathbf{h}_{t-1} - \mathbf{h}^e) + \mathbf{J}^{\text{inp}}\big|_{(\mathbf{h}^e, \mathbf{x}^e)} (\mathbf{x}_t - \mathbf{x}^e), \tag{2}$$

where we have defined the recurrent and input Jacobians as $J_{ij}^{\text{rec}}(\mathbf{h}, \mathbf{x}) := \frac{\partial F(\mathbf{h}, \mathbf{x})_i}{\partial h_j}$ and $J_{ij}^{\text{inp}}(\mathbf{h}, \mathbf{x}) := \frac{\partial F(\mathbf{h}, \mathbf{x})_i}{\partial x_j}$, respectively. If we expand about a fixed point $\mathbf{h}^* \approx F(\mathbf{h}^*, \mathbf{x} = \mathbf{0})$, the effect of an input $\mathbf{x}_t$ on the hidden state $\mathbf{h}_{T \geq t}$ can be approximated by $(\mathbf{J}^{\text{rec}})^{T-t} \mathbf{J}^{\text{inp}} \mathbf{x}_t$. Writing the eigendecomposition, $\mathbf{J}^{\text{rec}} = \mathbf{R} \boldsymbol{\Lambda} \mathbf{L}$, with $\mathbf{L} = \mathbf{R}^{-1}$, we have

$$(\mathbf{J}^{\text{rec}})^{T-t} \mathbf{J}^{\text{inp}} \mathbf{x}_t = \mathbf{R} \boldsymbol{\Lambda}^{T-t} \mathbf{L} \mathbf{J}^{\text{inp}} \mathbf{x}_t = \sum_{a=1}^{n} \mathbf{r}_a \lambda_a^{T-t} \ell_a^\top \mathbf{J}^{\text{inp}} \mathbf{x}_t, \tag{3}$$

where $\boldsymbol{\Lambda}$ is the diagonal matrix containing the (complex) eigenvalues, $\lambda_1 \geq \lambda_2 \geq \cdots \geq \lambda_n$ that are sorted in order of decreasing magnitude; $\mathbf{r}_a$ are the columns of $\mathbf{R}$; and $\ell_a^\top$ are the rows of $\mathbf{L}$. The magnitude of the eigenvalues of $\mathbf{J}^{\text{rec}}$ correspond to a time constant $\tau_a = \left| \frac{1}{\log|\lambda_a|} \right|$. The time constants, $\tau_a$, approximately determine how long and what information the system remembers from a given input.

We find fixed points by minimizing a function which computes the magnitude of the displacement $F(\mathbf{h}, \mathbf{x} = \mathbf{0}) - \mathbf{h}$ resulting from applying the update rule at point $\mathbf{h}$. That is, we numerically solve

$$\min_{\mathbf{h}} \frac{1}{2} \|\mathbf{h} - F(\mathbf{h}, \mathbf{x} = \mathbf{0})\|_2^2. \tag{4}$$

We seed the minimization procedure with hidden states visited by the network while processing test examples. To better sample the region, we also add some isotropic Gaussian noise to the initial points.

## A.2 Dimensionality Measures

Here we provide details regarding the measures used to determine both the dimensionality of our hidden-state and fixed-point manifolds. When we discuss the dimensionality of a set of points, we will mean their *intrinsic dimensionality*. Roughly, this is the dimensionality of a manifold that summarizes the discrete data points, accounting for the fact said manifold could be embedded in a higher-dimensional space in a non-linear fashion. For example, if the discrete points lie along a one-dimensional line that is non-linearly embedded in some two-dimensional space, then the measure of intrinsic dimensionality should be close to 1.

Let $X = \{X_1, \ldots, X_M\}$ be the set of $M$ points $X_I$ for $I = 1, \ldots, M$ for which we wish to measure the dimensionality. In this text, $X$ is either a set of hidden-states or a set of fixed points and each $X_I \in \mathbb{R}^n$ is a point in hidden-state space. To determine an accurate measure of dimensionality, we use the following measures:

- **Variance explained threshold**. Let $\mu_1 \geq \mu_2 \geq \ldots \geq \mu_n$ be the eigenvalues from PCA (i.e. the variances) on $X$. A simple measure of dimensionality is to threshold the number of PCA dimensions needed to reach a certain percentage of variance explained. For low number of classes, this threshold can simply be set at fixed values 90% or 95%. However, we would expect such threshold to breakdown as the number of classes increase, so we also use an $N$-dependent threshold of $N/(N+1)\%$.

- **Global participation ratio**. Again using PCA on $X$ as above, the participation ratio (PR) is defined to be a scalar function of the eigenvalues:

$$\text{PR} := \frac{\left(\sum_{i=1}^{n} \mu_i\right)^2}{\sum_{i=1}^{n} \mu_i^2}. \tag{5}$$

Intuitively, this is a scalar measure of the number of "important" PCA dimensions.

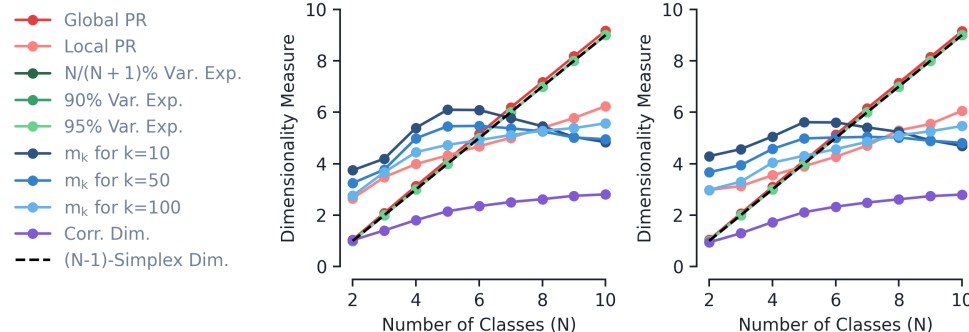

Figure 6: Various dimensionality measures as a function of number of classes, $N$, for the categorical synthetic data. Specifically, **Left:** the hidden state space and **Right:** the fixed point space dimensionality. This is for an $\ell_2$ of $5 \times 10^{-4}$ and each datapoint is an average over 10 initializations. The dotted black line shows the predicted dimensionality of a regular $(N-1)$-simplex.

- **Local participation ratio**. Since PCA is a linear mapping, both of the above measures will fail if the manifold is highly non-linear. We thus implement a local PCA as follows: we choose a random point and compute its $k$ nearest neighbors, then perform PCA on this subset of $k+1$ points. We then calculate the participation ratio on the eigenvalues of the local PCA using equation 5. We repeat the process over several random points, and then average the results. This measure is dependent upon the hyperparameter $k$.

- **MLE measure of intrinsic dimension** (Levina & Bickel, 2005). This is a nearest-neighbor based measure of dimension. For a point $X_I$, let $T_k(X_I)$ be the Euclidean distance to its $k$th nearest neighbor. Define the scalar quantities

$$\hat{m}_k = \frac{1}{M} \sum_{I=1}^{M} \hat{m}_k(X_I), \qquad \hat{m}_k(X_I) = \left[ \frac{1}{k-1} \sum_{j=1}^{k-1} \log \frac{T_k(X_I)}{T_j(X_I)} \right]^{-1}. \tag{6}$$

This measure is also dependent upon the number of nearest neighbors $k$.

- **Correlation Dimension** (Procaccia & Grassberger, 1983; Camastra & Vinciarelli, 2002). Define the scalar quantity

$$C_N(r) = \frac{2}{N(N-1)} \sum_{I=1}^{N} \sum_{J=I+1}^{N} \mathbf{1}\{\|X_I - X_J\|_2 < r\}. \tag{7}$$

Then, plotting $\log C_N(r)$ as a function of $\log r$, the correlation dimension is found by estimating the slope of the linear part of the plot.

We plot these dimensionality measures used on synthetic categorical data for class sizes $N = 2$ to $10$ in Figure 6. Despite their simplicity, we find the $95\%$ variance explained threshold and the global participation ratio to be the best match to what is theoretically predicted, hence we use these measures in the main text and in what follows.

## B  MODELS AND TRAINING

The three architectures we study are specified below, with $\mathbf{W}$ and $\mathbf{b}$ respectively representing trainable weight matrices and bias parameters, and $\mathbf{h}_t$ denoting the hidden state at timestep $t$. All other vectors ($\mathbf{c}, \mathbf{g}, \mathbf{r}, \mathbf{i}, \mathbf{f}$) represent intermediate quantities; $\sigma(\cdot)$ represents a pointwise sigmoid nonlinearity; and $f(\cdot)$ is the tanh nonlinearity.

**Update-Gate RNN (UGRNN)**

$$\mathbf{h}_t = \mathbf{g} \cdot \mathbf{h}_{t-1} + (1 - \mathbf{g}) \cdot \mathbf{c}, \qquad \begin{aligned} \mathbf{c} &= f\left(\mathbf{W}^{\mathrm{ch}}\mathbf{h}_{t-1} + \mathbf{W}^{\mathrm{cx}}\mathbf{x}_t + \mathbf{b}^{\mathrm{c}}\right), \\ \mathbf{g} &= \sigma\left(\mathbf{W}^{\mathrm{gh}}\mathbf{h}_{t-1} + \mathbf{W}^{\mathrm{gx}}\mathbf{x}_t + \mathbf{b}^{\mathrm{g}}\right). \end{aligned} \tag{8}$$

**Gated Recurrent Unit (GRU)**

$$\mathbf{h}_t = \mathbf{g} \cdot \mathbf{h}_{t-1} + (1 - \mathbf{g}) \cdot \mathbf{c}, \qquad \begin{aligned} \mathbf{c} &= f\left(\mathbf{W}^{\mathrm{ch}}(\mathbf{r} \cdot \mathbf{h}_{t-1}) + \mathbf{W}^{\mathrm{cx}}\mathbf{x}_t + \mathbf{b}^{\mathrm{c}}\right), \\ \mathbf{g} &= \sigma\left(\mathbf{W}^{\mathrm{gh}}\mathbf{h}_{t-1} + \mathbf{W}^{\mathrm{gx}}\mathbf{x}_t + \mathbf{b}^{\mathrm{g}}\right), \\ \mathbf{r} &= \sigma\left(\mathbf{W}^{\mathrm{rh}}\mathbf{h}_{t-1} + \mathbf{W}^{\mathrm{rx}}\mathbf{x}_t + \mathbf{b}^{\mathrm{r}}\right). \end{aligned} \qquad (9)$$

**Long-Short-Term-Memory (LSTM)**

$$\mathbf{h}_t = \begin{bmatrix} \mathbf{c}_t \\ \widetilde{\mathbf{h}}_t \end{bmatrix}, \qquad \begin{aligned} \widetilde{\mathbf{h}}_t &= f(\mathbf{c}_t) \cdot \sigma\left(\mathbf{W}^{\mathrm{hh}}\mathbf{h}_{t-1} + \mathbf{W}^{\mathrm{hx}}\mathbf{x}_t + \mathbf{b}^{\mathrm{h}}\right), \\ \mathbf{c}_t &= \mathbf{f}_t \cdot \mathbf{c}_{t-1} + \mathbf{i}_t \cdot \sigma\left(\mathbf{W}^{\mathrm{ch}}\widetilde{\mathbf{h}}_{t-1} + \mathbf{W}^{\mathrm{cx}}\mathbf{x}_t + \mathbf{b}^{\mathrm{c}}\right), \\ \mathbf{i}_t &= \sigma\left(\mathbf{W}^{\mathrm{ih}}\mathbf{h}_{t-1} + \mathbf{W}^{\mathrm{ix}}\mathbf{x}_t + \mathbf{b}^{\mathrm{i}}\right), \\ \mathbf{f}_t &= \sigma\left(\mathbf{W}^{\mathrm{fh}}\mathbf{h}_{t-1} + \mathbf{W}^{\mathrm{fx}}\mathbf{x}_t + \mathbf{b}^{\mathrm{f}}\right). \end{aligned} \qquad (10)$$

With the natural datasets, we form the input vectors $\mathbf{x}_t$ by using a (learned) 128-dimensional embedding layer. These UGRNNs and GRUs have hidden-state dimension $n = 256$, while in the LSTMs, both the hidden-state $\widetilde{\mathbf{h}}_t$ and the memory $\widetilde{\mathbf{c}}_t$ are 256-dimensional, yielding a total hidden-state dimension $n = 512$. For the synthetic datasets, due to their small vocabulary size, we simply pass one-hot encoded inputs in the RNN architectures, i.e. we use no embedding layer. For UGRNNs and GRU, we use a hidden-state dimension of $n = 128$, while for LSTMs we again use the same dimension for both $\widetilde{\mathbf{h}}_t$ and $\widetilde{\mathbf{c}}_t$, resulting in a total hidden-state dimension of $n = 256$.

The model's predictions (logits for each class) are computed by passing the final hidden state $\mathbf{h}_T$ through a linear layer. In the synthetic experiments, we do not add a bias term to this linear readout layer, chosen for simplicity and ease of interpretation.

We train the networks using the ADAM optimizer (Kingma & Ba, 2014) with an exponentially-decaying learning rate schedule. We train using cross-entropy loss with added $\ell_2$ regularization, penalizing the squared $\ell_2$ norm of the network parameters. Natural experiments use a batch size of 64 with initial learning rate $\eta = 0.01$, clipping gradients to a maximum value of 30; the learning rate decays by $0.9984$ every step. Synthetic experiments use a batch size of 128, initial learning rate $\eta = 0.1$, and a gradient clip of 10; the learning rate decays by $0.9997$ every step.

## C  NATURAL DATASET DETAILS

We use the following text classification datasets in this study:

- The **Yelp reviews dataset** (Zhang et al., 2015) consists of Yelp reviews, labeled by the corresponding star rating (1 through 5). Each of the five classes features 130,000 training examples and 10,000 test examples. The mean length of a review is 143 words.

- The **Amazon reviews dataset** (Zhang et al., 2015) consists of reviews of products bought on Amazon.com over an 18-year period. As with the Yelp dataset, these reviews are labeled by the corresponding star rating (1 through 5). Each of the five classes features 600,000 training examples and 130,000 test examples. The mean length of a review is 86 words.

- The **DBPedia ontology dataset** (Zhang et al., 2015) consists of titles and abstracts of Wikipedia articles in one of 14 non-overlapping categories, from DBPedia 2014. Categories include: company, educational institution, artist, athlete, office holder, mean of transportation, building, natural place, village, animal, plant, album, film, and written work. Each class contains 40,000 training examples and 5,000 testing examples. We use the abstract only for classification; mean abstract length is 56 words.

- The **AG's news corpus** (Zhang et al., 2015) contains titles and descriptions of news articles from the web, in the categories: world, sports, business, sci/tech. Each category features 30,000 training examples and 1,900 testing examples. We use only the descriptions for classification; the mean length of a description is 35 words.

- The **GoEmotions dataset** (Demszky et al., 2020) contains text from 58,000 Reddit comments collected between 2005 and 2019. These comments are labeled with the following 27

emotions: admiration, approval, annoyance, gratitude, disapproval, amusement, curiosity, love, optimism, disappointment, joy, realization, anger, sadness, confusion, caring, excitement, surprise, disgust, desire, fear, remorse, embarrassment, nervousness, pride, relief, grief. The mean length of a comment is 16 words.

Two main characteristics distinguish these datasets: (i) whether there is a notion of *order* among the class labels, and (ii) whether labels are exclusive. The reviews datasets, Amazon and Yelp, are naturally ordered, while the labels in the other datasets are unordered. All of the datasets besides GoEmotions feature exclusive labels; only in GoEmotions can two or more labels (e.g., the emotions *anger* and *disappointment*) characterize the same example. In addition to the standard five-class versions of the ordered datasets, we form three-class subsets by collecting reviews with 1, 3, and 5 stars (excluding reviews with 2 and 4 stars).

We build a vocabulary for each dataset by converting all characters to lowercase and extracting the 32,768 most common words in the training corpus. Tokenization is done by TensorFlow TF.Text WordpieceTokenizer.

## D  SYNTHETIC DATASET DETAILS

In this appendix we provide many additional details and results from our synthetic datasets. Although these datasets represent significantly simplified settings compared to their realistic counterparts, often the results from training RNNs on the synthetic and natural datasets are strikingly similar.

### D.1  CATEGORICAL DATASET

For the categorical synthetic dataset used in Section 3.1, we generate phrases of $L$ words, drawing from a word bank consisting of $N + 1$ words, $\mathcal{W} = \{\text{evid}_1, \ldots, \text{evid}_N, \text{neutral}\}$. Each word $W \in \mathcal{W}$ has an $N$-dimensional vector of integers associated with it, $\mathbf{w}^W = \{w_1^W, \ldots, w_N^W\}$ with $w_i^W \in \mathbb{Z}$ for all $i = 1, \ldots, N$. The word "evid$_i$" has score defined by $w_i^{\text{evid}_i} = 1$ and $w_j^{\text{evid}_i} = 0$ for $i \neq j$. Meanwhile, the word "neutral" has $w_j^{\text{neutral}} = 0$ for all $j$. Additionally, each phrase has a corresponding score $\mathbf{s}$ that also takes the form of an $N$-dimensional vector of integers, $\mathbf{s} = \{s_1, \ldots, s_N\}$. A phrase's score is equal to the sum of scores of the words contained in said phrase, $\mathbf{s} = \sum_{W \in \text{phrase}} \mathbf{w}^W$. The phrase is then assigned a label corresponding to the class with the maximum score, $y = \text{argmax}(\mathbf{s})$.[5]

In the main text we analyze synthetic datasets where phrases are drawn from a uniform distribution over all possible scores, $\mathbf{s}$. To do so, we enumerate all possible scores a phrase of length $L$ can produce as well as all possible word combinations that can generate a given score. It is also possible to build phrases by drawing each word from a uniform distribution over all words in $\mathcal{W}$. In practice, we find all results on synthetic datasets have minor quantitative differences when comparing these two methods, but qualitatively the results are the same.[6]

As highlighted in the main text, after training on this synthetic data we find the explored hidden-state space to resemble a regular $(N-1)$-simplex. This holds for a large range $\ell_2$ values relative to the natural datasets. In Figure 7, we plot the (global) participation ratio, defined in equation 5, as a function of the number of classes, $N$.

In addition to the hidden states forming a simplex, we observe the $N$ readout vectors are approximately equal magnitude and are aligned along the $N$ vertices of said $(N-1)$-simplex. In Figure 8, we plot several measures on the readout vectors that support this claim that we now discuss. We find the readout vectors to have very close to the same magnitude (Fig. 8, left panel). The angle (in degrees) between a pair of vectors that point from the center of a $(N-1)$-simplex to two of its

---

[5]For phrases with multiple occurrences of the maximum score, the phrase is labeled by the class with the smallest numerical index.

[6]We have also analyzed the same synthetically generated data for variable phrase lengths. The qualitative results focused on in this text did not change in this setting.

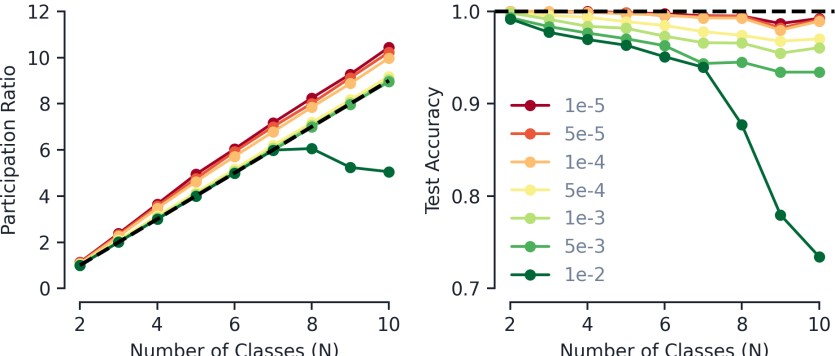

Figure 7: How the dimensionality and accuracy of categorical synthetic data changes as a function of $\ell_2$ regularization. Each datapoint is an average over 10 initializations. **Left:** Hidden state space dimensionality (global participation ratio) and **Right:** test accuracy as a function of number of classes, $N$, for the unordered synthetic data for several different values of $\ell_2$. The dotted black line shows the predicted regular $(N-1)$-simplex dimensionality.

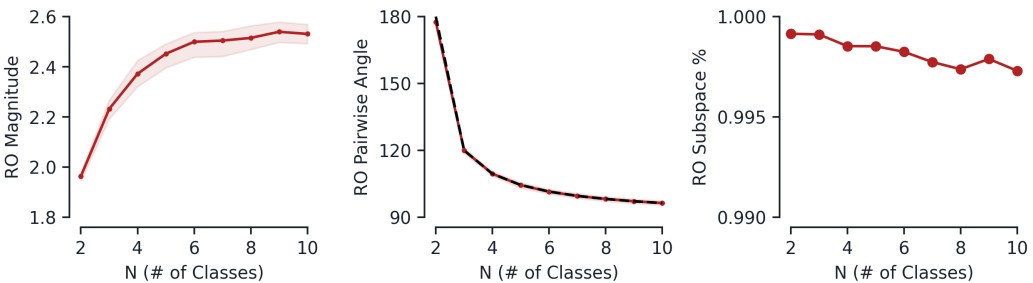

Figure 8: Readout (RO) measures as a function of the number of classes, $N$ for synthetic categorical data. All data collected over 10 random initializations at an $\ell_2$ of $5 \times 10^{-4}$. **Left:** Magnitude, with the line being the median and filled in area being 10th and 90th percentile. **Center:** Pairwise angle, again with median being the solid line with filled in 10th and 90th percentiles (the variance in angles is very low, so this is almost indistinguishable from the line itself). Black dotted line is theoretical simplex angle. **Right:** Subspace percentage, $\Lambda$, as defined in equation 12.

vertices is

$$\theta_{\text{theory}} = \frac{180}{\pi} \times \arccos\left(-\frac{1}{N-1}\right) . \tag{11}$$

For example, for a regular 2-simplex, i.e. an equilateral triangle, this predicts an angle between readout vectors of 120 degrees. The distribution of pairwise angles between readout vectors is plotted the center panel of Figure 8. Lastly, if the readouts lie entirely within the $(N-1)$-simplex, all $N$ of them should live in the same $\mathbb{R}^{N-1}$ subspace. To measure this, define $\mathbf{r}'_i$ to be projection of $\mathbf{r}_i$ into the subspace formed by the other $N-1$ readout vectors, i.e. the span of the set $\{\mathbf{r}_j \,|\, j = 1, \ldots, N; j \neq i\}$. We then define the subspace percentage, $\Lambda$, as follows,

$$\Lambda := \frac{1}{N} \sum_{i=1}^{N} \frac{\|\mathbf{r}'_i\|_2}{\|\mathbf{r}_i\|_2} . \tag{12}$$

If all the readouts lie within the same $\mathbb{R}^{N-1}$ subspace, then $\Lambda = 1$. The right panel of Figure 8 shows that in practice $\Lambda \approx 1$ for the synthetic data with $\ell_2$ regularization parameter of $5 \times 10^{-4}$.

**Why a Regular $(N-1)$-Simplex?** Here we propose an intuitive scenario that leads the network's hidden states to form a regular $(N-1)$-simplex. To classify a given phrase correctly, the network must learn to keep track of the value of the $N$-dimensional score vector $\mathbf{s}$. One way this can be

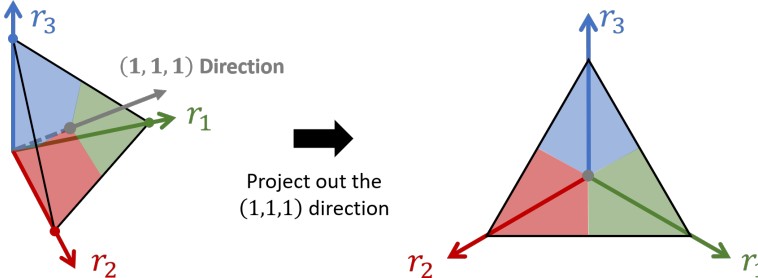

Figure 9: **Left:** Full space of possible scores as a subspace of $\mathbb{R}^3$. **Right:** Two-dimensional space resulting from projecting out the $(1, 1, 1)$ direction of the $\mathbb{R}^3$ subspace, forming a regular 2-simplex.

done is follows: Let the network's hidden state live in some $\mathbb{R}^N$ dimensional subspace. Within this subspace, let the $N$ readout vectors be orthogonal and have equal magnitude. Furthermore, define a Cartesian coordinate system to have basis vectors aligned with the $N$ readouts, with $z_i$ the coordinate along the direction of readout $\mathbf{r}_i$. Then, the coordinates within this subspace encodes the components of the $N$-dimensional score vector $\mathbf{s}$: the evidence word 'evid$_i$ moves you along the coordinate direction $i$ some fixed amount and so $s_i \propto z_i$. Note the subspace of $\mathbb{R}^N$ explored by hidden states has a finite extent, since the phrases are of finite length. This subspace can be further subdivided into regions corresponding to different class labels: if $z_i > z_j$ for all $j \neq i$ then $\mathbf{h} \cdot \mathbf{r}_i > \mathbf{h} \cdot \mathbf{r}_j$ and the phrase is classified as Class $i$. The left panel of Figure 9 shows an example of the 3-dimensional subspace for $N = 3$.

The important step that gets us from a subspace of $\mathbb{R}^N$ to the regular $(N-1)$-simplex is the presence of the softmax layer used when calculating loss. Since this function normalizes the scores, it is only the *relative* size of the components of $\mathbf{s}$ that matters. Removing the dependence on the absolute score values corresponds to projecting onto the $\mathbb{R}^{N-1}$ subspace orthogonal to the $N$-dimensional ones vector, $(1, 1, \ldots, 1)$. This projection results in an $(N-1)$-simplex with the readouts aligned with the vertices. A demonstration of this procedure for $N = 3$ is shown in Figure 9.

## D.2 Ordered Dataset

As alluded to in the main text, we try two renditions of ordered synthetic data. The details of both are given below. The first relies on a ground truth of only a sentiment score, while the second classifies based on both sentiment and neutrality. Although the first is simpler and still bears many resemblances to natural data (i.e. Yelp and Amazon), we find the second to be a better match overall.

**Sentiment Only Synthetic Data** The first synthetic data for ordered datasets is very similar into that of the categorical sets with a minor difference in the word bank and how phrases are assigned labels. For $N$-class ordered datasets, the word bank always consists of only three words $\mathcal{W} = \{\text{good}, \text{bad}, \text{neutral}\}$. We now take the word and phrase scores to be 1-dimensional and $w_1^{\text{good}} = +1$, $w_1^{\text{bad}} = -1$, and $w_1^{\text{neutral}} = 0$. We then subdivide the range of possible scores $s$ into $N$ equal regions, and a phrase is labeled by the region which its score fall into. Given the above definitions, the range of scores is $[-L, L]$, and so for the $N = 3$, with labels $\{\text{Positive}, \text{Negative}, \text{Neutral}\}$, we define some threshold $s_{\text{N}} = L/3$. Then a label $y$ is assigned as follows:

$$y = \begin{cases} \text{Positive} & s \geq s_{\text{N}}, \\ \text{Neutral} & |s| < s_{\text{N}}, \\ \text{Negative} & s \leq -s_{\text{N}}, \end{cases} \tag{13}$$

Meanwhile, for $N = 5$, one could draw the region divisions at the score values $\{-3L/5, -L/5, L/5, 3L/5\}$. Similar to the categorical data above, in the main text we draw phrases from a uniform distribution over all possible scores.

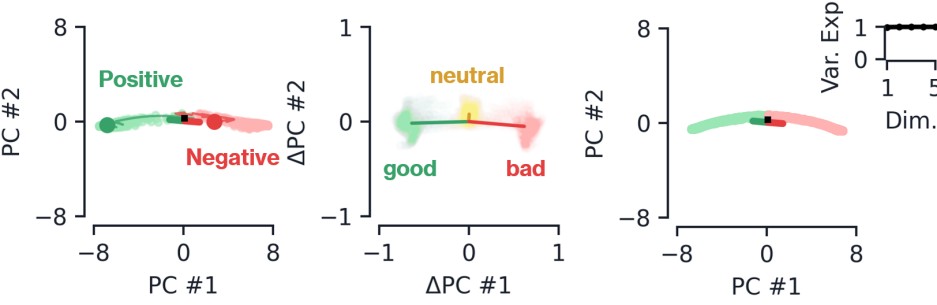

Figure 10: Synthetic ordered data for $N = 2$. **Left:** Final hidden states for 600 test samples, colored by their label, with a few example hidden states trajectories shown explicitly. The initial state $\mathbf{h}_0$ is shown as a black square, and the three thick solid lines are the three readouts, colored by their respective class. **Center:** The hidden deflections from the three words in the vocabulary, with the average deflection of each shown as a solid line. **Right:** Fixed points, colored by their predicted label. The inset shows the variance explained.

The $N = 2$ case corresponds to sentiment analysis, and its hidden state space, word deflections, and fixed point manifold are plotted in Figure 10.[7] The dynamics of this system qualitatively match the natural dataset analyzed in Maheswaranathan et al. (2019). Briefly, the sentiment score is encoded in the hidden state's position along a one-dimensional line, aligned with the readouts which point in opposite directions. The word 'good' ('bad') moves you along this line along the 'Positive' ('Negative') readout, increasing the corresponding logit value.

The simplest ordered dataset beyond binary sentiment analysis is that of $N = 3$, and a plot showing the final hidden states, deflections, and fixed-point manifold is shown in top row of Figure 11. In the bottom row, we show the same plots for $N = 5$. In both cases, the hidden-state trajectories move away from $\mathbf{h}_0$ onto a curve embedded in 2d plane, with the curve bent around the origin of said plane. The $N$ readout vectors are evenly fanned out in the 2d plane, which subdivides the curve into $N$ regions corresponding to each of the $N$ classes. The curve subdivisions reflect the ordering of the score subdivisions, for $N = 3$ we see 'Neutral' lying in between 'Positive' and 'Negative' and for $N = 5$ the stars are ordered from 1 to 5.

In contrast to categorical data, the word deflection $\Delta \mathbf{h}_t$ are highly varied and have a strong dependence on a state's location in hidden-state space. On average, the words 'good' and 'bad' move the hidden state further left/right along the curve. Although $\Delta \mathbf{h}_t$ for the word 'neutral' is on average smaller, it tends to move the hidden state along the 'Neutral' or '3 Star' readout. These dynamics are how the network encodes the relative count of 'good' and 'bad' words in a phrase that ultimately determines the phrase's classification. We show the fixed points in the far right panel of Figure 11. For $N = 3$, the fixed point manifold mostly resembles that of a one-dimensional bent line attractor, with a small region that is two-dimensional along the 'Neutral' readout. For $N = 5$, the fixed point manifold is much more planar. Thus, the $N = 3$ case exhibits very similar dynamics to that of the line attractor studied in Maheswaranathan et al. (2019), the attractor is now simply subdivided into three regions due to the readout vector alignments.

**Sentiment and Neutrality Synthetic Data** Instead of classifying a phrase based off a single sentiment score, our second ordered synthetic model classifies a phrase based off of two scores that track the sentiment and intensity of a given phrase. We draw from an enhanced word bank consisting of $\mathcal{W} = \{$awesome, good, okay, bad, awful, neutral$\}$. We take the two-dimensional word score to have components corresponding to (sentiment, intensity) where positive (negative) sentiment scores correspond to positive (negative) sentiment and positive (negative) intensity scores correspond to high

---

[7]The $N = 2$ ordered dataset is equivalent to the $N = 2$ categorical dataset. Intuitively, 'good' and 'bad' can be though of evidence vectors for the classes 'Positive' and 'Negative', respectively. Just like the categorical classification, whichever of these evidence words appears the most in a given phrase will be the phrase's label.

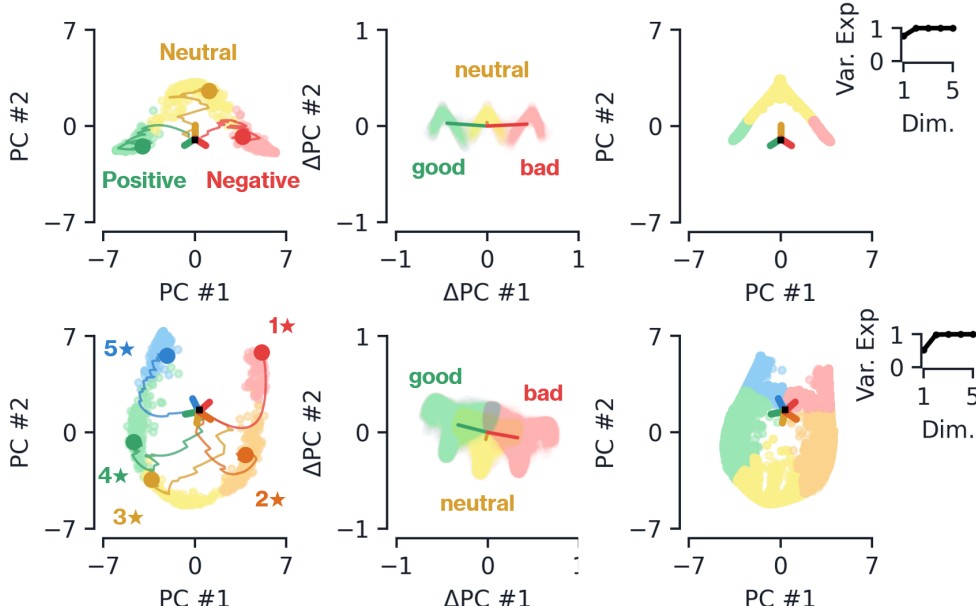

Figure 11: Synthetic ordered data for $N = 3$ (top row) and $N = 5$ (bottom row). **Left:** Final hidden states for 600 test samples, colored by their label, with a few example hidden states trajectories shown explicitly. The initial state $\mathbf{h}_0$ is shown as a black square, and the three thick solid lines are the three readouts, colored by their respective class. **Center:** The hidden deflections from the three words in the vocabulary, with the average deflection of each shown as a solid line. **Right:** Fixed points, colored by their predicted label. The inset shows the variance explained.

(low) emotion. The word score values we use are

$$\mathbf{w}^{\text{awesome}} = (2, 1), \qquad \mathbf{w}^{\text{good}} = (1, -1/2), \qquad \mathbf{w}^{\text{okay}} = (0, -2), \qquad (14a)$$

$$\mathbf{w}^{\text{bad}} = (-1, -1/2), \qquad \mathbf{w}^{\text{awful}} = (-2, 1), \qquad \mathbf{w}^{\text{neutral}} = (0, 0). \qquad (14b)$$

As with the other synthetic models, we sum all word scores across a phrase to arrive at a phrase's sentiment and intensity score, $(s, i)$. We then assign the phrase a label $y$ based off the following criterion:

$$y = \begin{cases} \text{Three Star} & i < 0 \text{ and } |i| > |s|, \text{ otherwise:} \\ \text{Five Star} & i \geq 0 \text{ and } s > 0, \\ \text{Four Star} & i < 0 \text{ and } s > 0, \\ \text{Two Star} & i < 0 \text{ and } s < 0, \\ \text{One Star} & i \geq 0 \text{ and } s \leq 0. \end{cases} \qquad (15)$$

Thus we see that scores with negative (low) intensity where the intensity magnitude is greater than the sentiment magnitude are classified as 'Three Star', i.e. it is a neutral phrase. Otherwise, phrases with low intensity that are the less extreme reviews are classified as either 'Two Star' or 'Four Star' based on their sentiment. Finally, phrases with high intensity are labeled either 'One Star' or 'Five Star', again based on their sentiment.

### D.3  MULTI-LABELED DATASET

Here we provide details of the synthetic multi-labeled dataset, that corresponds to natural dataset GoEmotions in Section 3.3 of the main text. Let us introduce this by taking the $N = 2$ as an explicit example, where each phrase can have up to two labels. We draw from a word bank consisting of $\mathcal{W} = \{\text{good}_1, \text{bad}_1, \text{good}_2, \text{bad}_2, \text{neutral}\}$, where

$$\mathbf{w}^{\text{neutral}} = (0, 0), \qquad \mathbf{w}^{\text{good}_1} = (1, 0), \qquad \mathbf{w}^{\text{bad}_1} = (-1, 0), \qquad (16a)$$

$$\mathbf{w}^{\text{good}_2} = (0, 1), \qquad \mathbf{w}^{\text{bad}_2} = (0, -1). \qquad (16b)$$

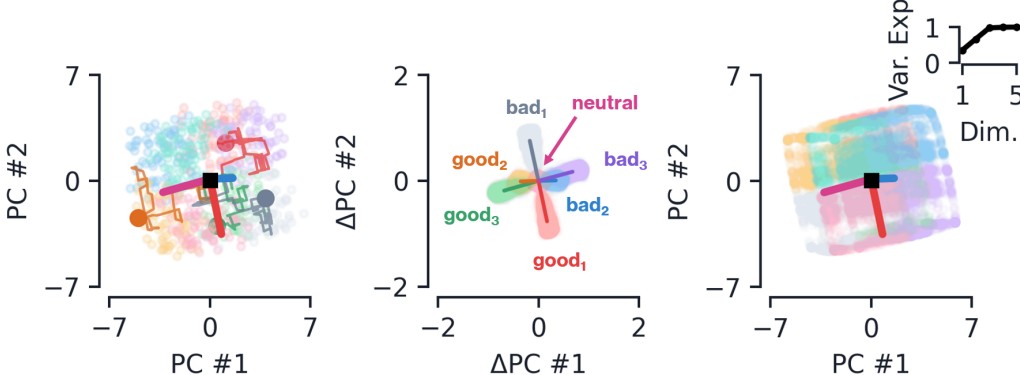

Figure 12: Synthetic multi-labeled example with $N = 3$. **Left:** A few example hidden states trajectories. Inset: variance explained over 600 hidden state examples. **Center:** Deflections from a given word input. **Right:** Fixed points and variance explained over fixed points.

We then classify each phrase with *two* labels, individually based on the score vector components $s_1$ and $s_2$. Namely,

$$y_1 = \begin{cases} \text{Positive}_1 & s_1 \geq 0, \\ \text{Negative}_1 & s_1 < 0, \end{cases} \qquad y_2 = \begin{cases} \text{Positive}_2 & s_2 \geq 0, \\ \text{Negative}_2 & s_2 < 0. \end{cases} \tag{17}$$

Thus there are four possible combinations of labels. For this synthetic datasets, we generate phrases by uniformly drawing words one-by-one from $\mathcal{W}$. Generalization of the above construction to an arbitrary number of possible labels $N$ is straightforward: one simply adds additional $N$-dimension score vectors $\mathbf{w}^{\text{good}_i}$ and $\mathbf{w}^{\text{bad}_i}$ for each possible label $i = 1, \ldots, N$ and then uses the $N$ components of the score to assign the $N$ labels, $y_i$, individually.

The results after training a network on the $N = 2$ dataset are shown in the main text in Figure 5, and results for $N = 3$ are shown in Figure 12. Again, we see the explored hidden-state space to be low-dimensional, but notably it now resembles a three-dimensional cube. This is certainly a large departure from the $N = 8$ categorical dataset, from which we expect a (seven-dimensional) regular 7-simplex. Instead, what we see here is the "outer product" of three $N = 2$ ordered datasets. That is, we expect a single $N = 2$ ordered dataset (i.e. binary sentiment analysis) to have a hidden-state space that resembles a line attractor. As one might expect, tasking the network with analyzing three such sentiments at once leads to three line attractors that are orthogonal to one another, forming a cube. This is supported in the center panel of Figure 5, where we see the various sentiment evidences are orthogonal from one another.

# E ADDITIONAL RESULTS ON NATURAL DATASETS

## E.1 AG NEWS

This subsection contains two figures: Figures 13 and 14 complement Figure 1 in the main text; the main text figure showed the manifolds learned by an LSTM on both 3- and 4-class AG News datasets; the figures in this appendix show corresponding manifolds learned by a GRU and UGRNN.

## E.2 DBPEDIA 3-CLASS AND 4-CLASS CATEGORICAL PREDICTION

Like AG News, DBPedia Ontology is a categorical classification dataset. We show results for networks trained on 3- and 4-class subsets of this dataset in Figures 15, 16, and 17.

## E.3 YELP 5-CLASS STAR PREDICTION

Figures 19, 18, and 20 show the fixed-point manifolds associated with a GRU, LSTM and UGRNN, respectively, trained on 5-class and 3-class Yelp dataset. These reviews are naturally five star; we

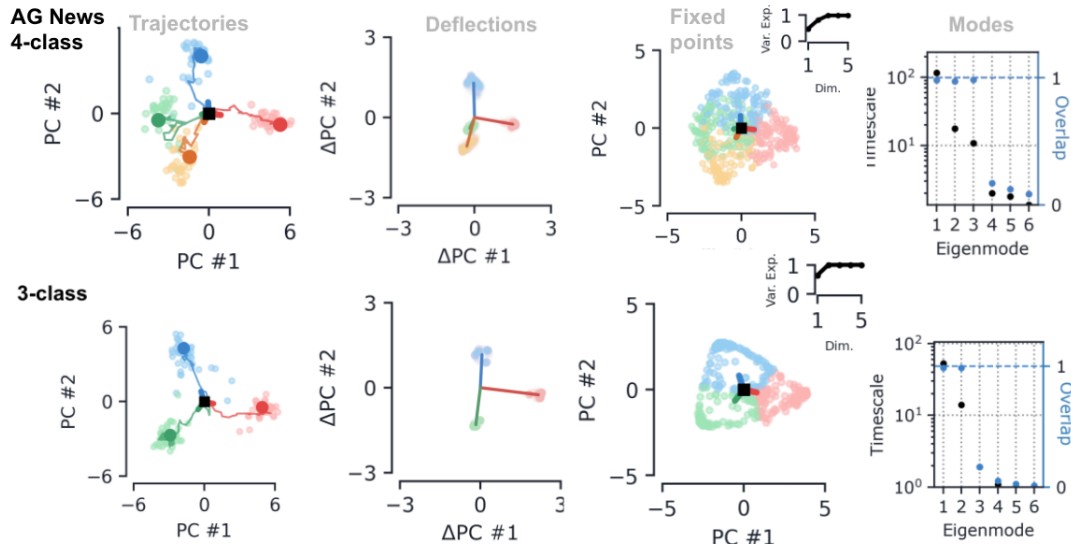

Figure 13: Trajectories, individual-word deflections, fixed-point manifolds, and eigenmodes learned by a LSTM on the 3- and 4-class AG News task. Solid lines in the trajectories and fixed-points plots show readout vectors. Note the similarity between this manifold and that learned by the GRU, described in the main text Figure 1. The words we use to generate the deflections are *sox*, *bankruptcy*, *military*, and *computer*.

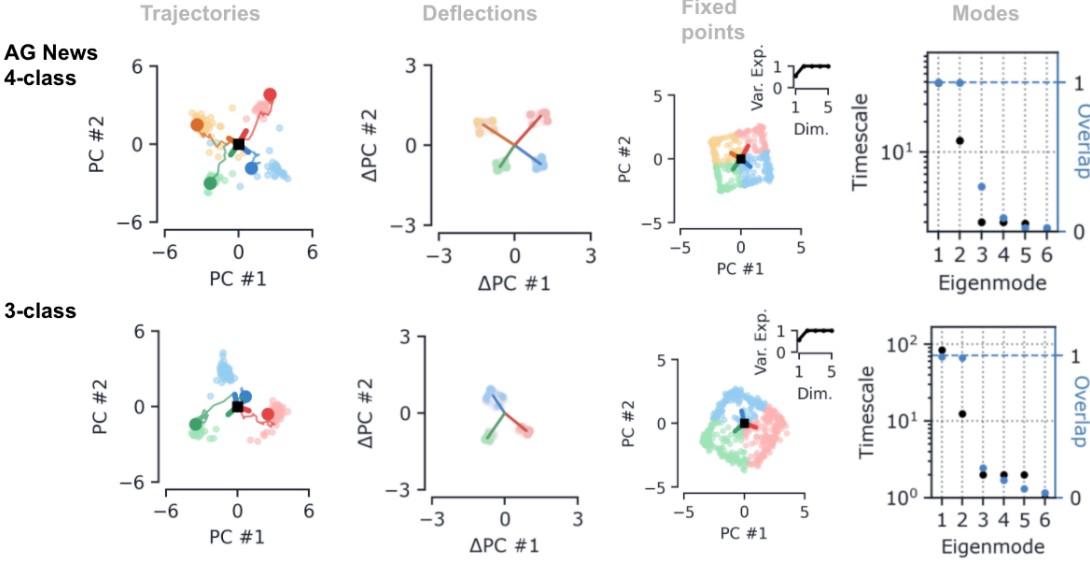

Figure 14: Trajectories, deflections, fixed-point manifolds, and eigenmodes learned by a UGRNN on 3- and 4-class AG News. Solid lines in the trajectories and fixed-points plots show readout vectors. Notice that in the 4-class case, unlike for GRUs or LSTMs, the UGRNN seems to always learn a square manifold. This is likely due to correlations in the input data (see Figure 29 for details).

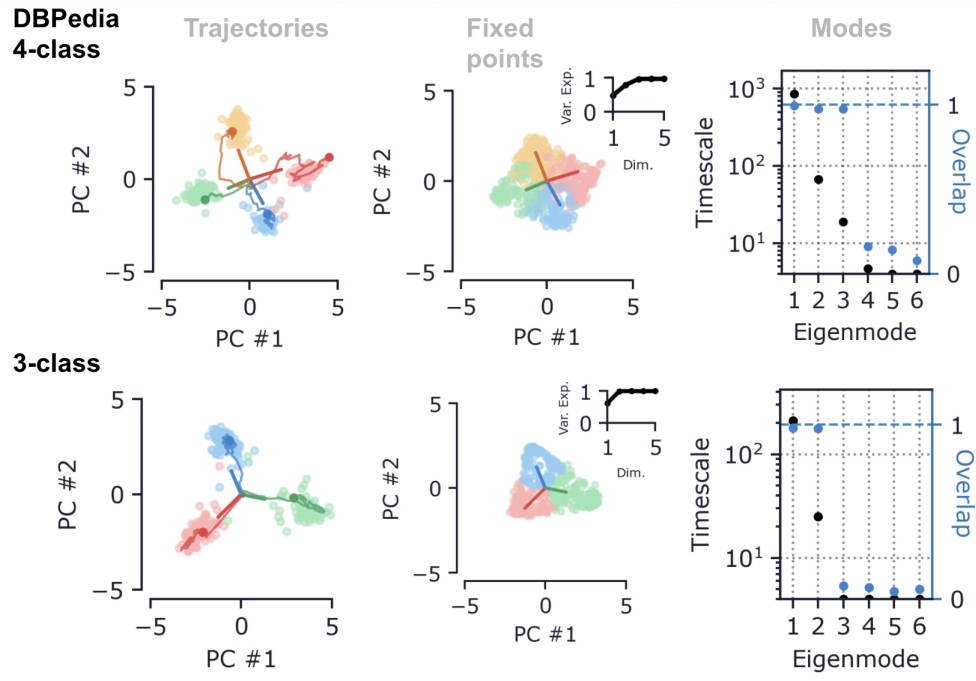

Figure 15: LSTM on four-class subset of the DBPedia ontology dataset

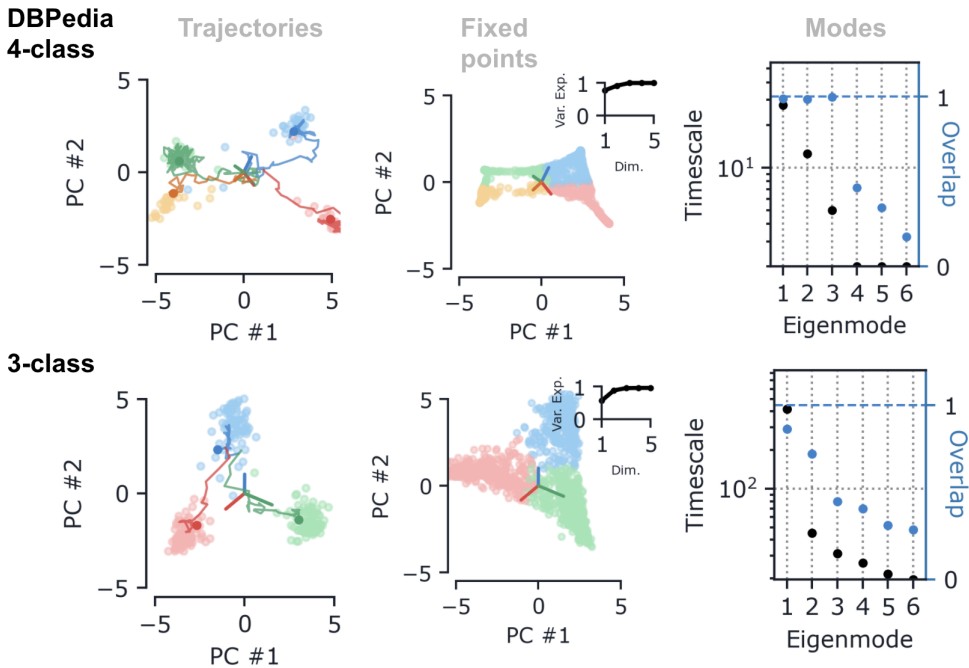

Figure 16: GRU on four-class subset of the DBPedia ontology dataset

create a 3-class subset by removing examples labeled with 2 and 4 stars. These figures complement Figure 1 in the main text.

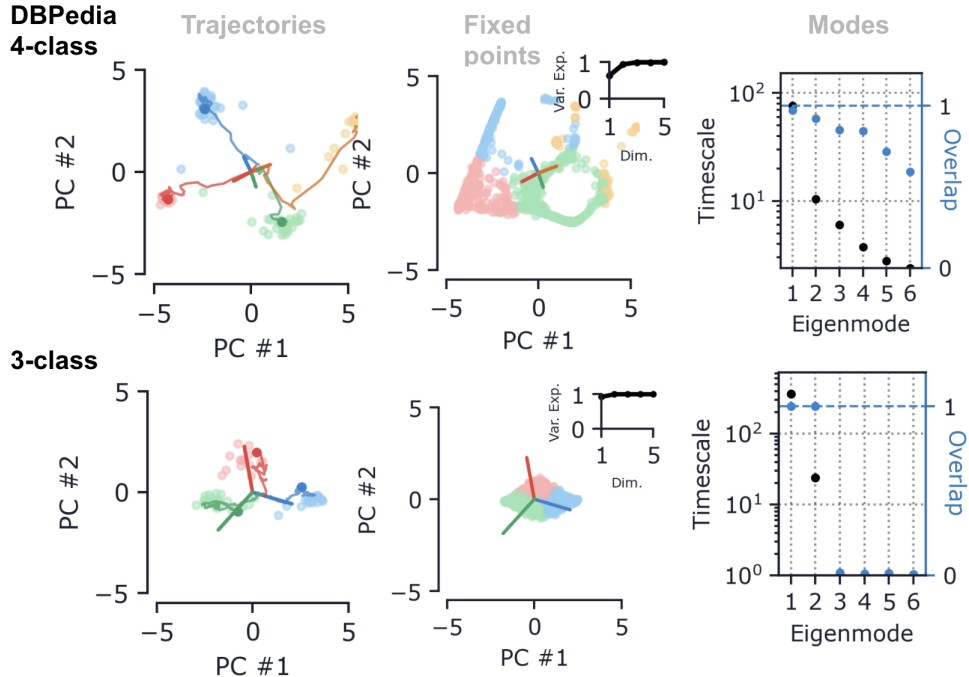

Figure 17: UGRNN on four-class subset of the DBPedia ontology dataset

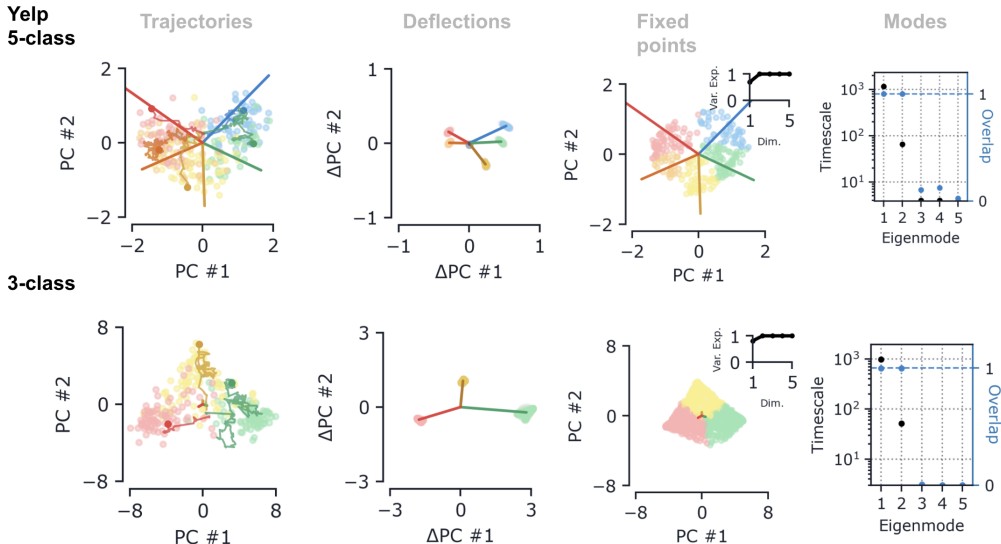

Figure 18: LSTM on five-class and three-class Yelp

## E.4 AMAZON 5-CLASS AND 3-CLASS STAR PREDICTION

As another example of an ordered dataset, Figures 21, 22, and 23 show results for networks trained on a 3-class and 5-class subsets of Amazon reviews. These reviews are naturally five star; we create a 3-class subset by removing examples labeled with 2 and 4 stars.

## E.5 3-CLASS GOEMOTIONS

In addition to the 2 class variant presented in the main text, we also trained a 3 class version of the GoEmotions dataset. We filtered the dataset to just include the following three classes: "admiration",

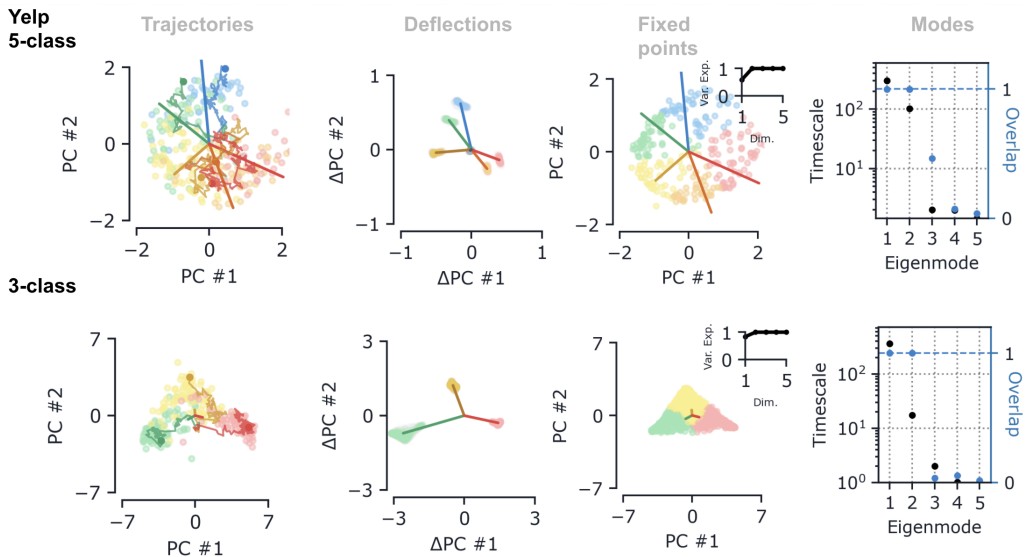

Figure 19: GRU on five-class and three-class Yelp

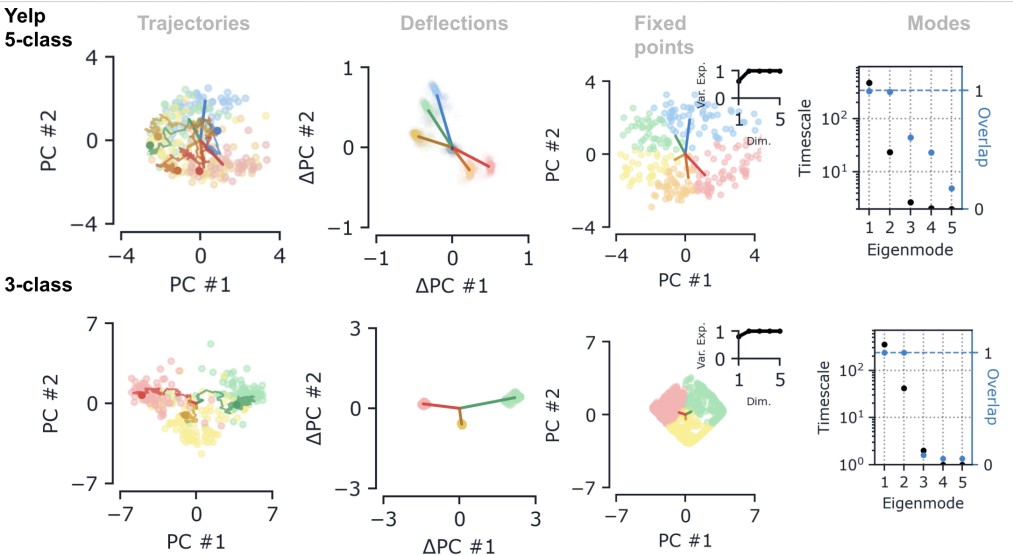

Figure 20: UGRNN on five-class and three-class Yelp

"approval", and "annoyance" (these were selected as they were the classes with the largest number of examples). These results are presented in Figure 24. For this network, despite having three classes, we find that the fixed points are largely two dimensional (Fig. 24a). The timescales of the eigenvalues of the Jacobian computed at these fixed points have two slow modes (Fig. 24b), which overlap with the two modes (Fig. 24c); thus we have a roughly 2D plane attractor. However, the participation ratio (Fig. 24d) indicates that the dimensionality of this attractor is slightly higher than the 2D case shown in Fig. 5. We suspect that these differences are due to the strong degree of class imbalance present in the GoEmotions dataset. There are very few examples with multiple labels, for any particular combination of labels. In synthetic multi-labeled data (which is class balanced), we see much clearer 3D structure when training a 3 class network (Fig. 12).

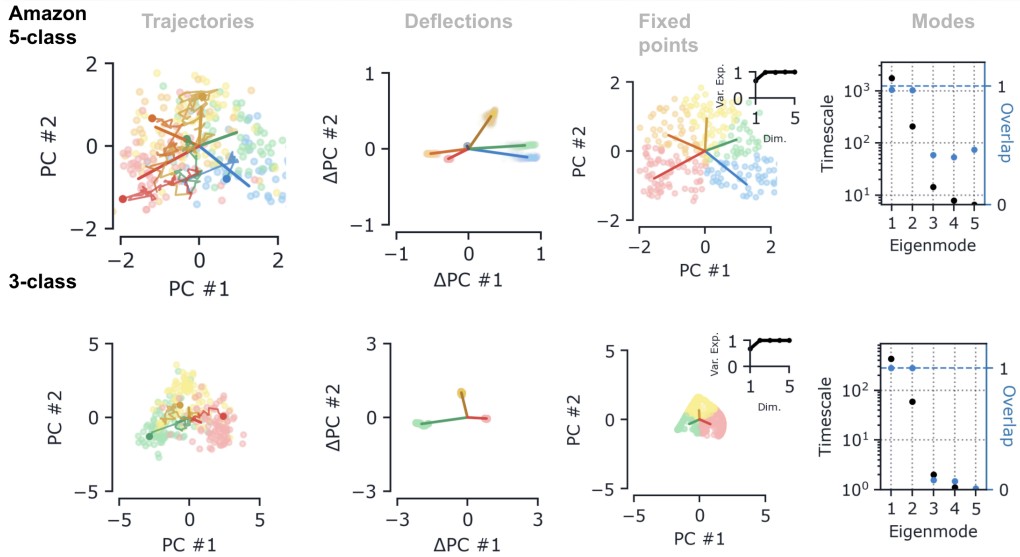

Figure 21: LSTM on five-class and three-class Amazon

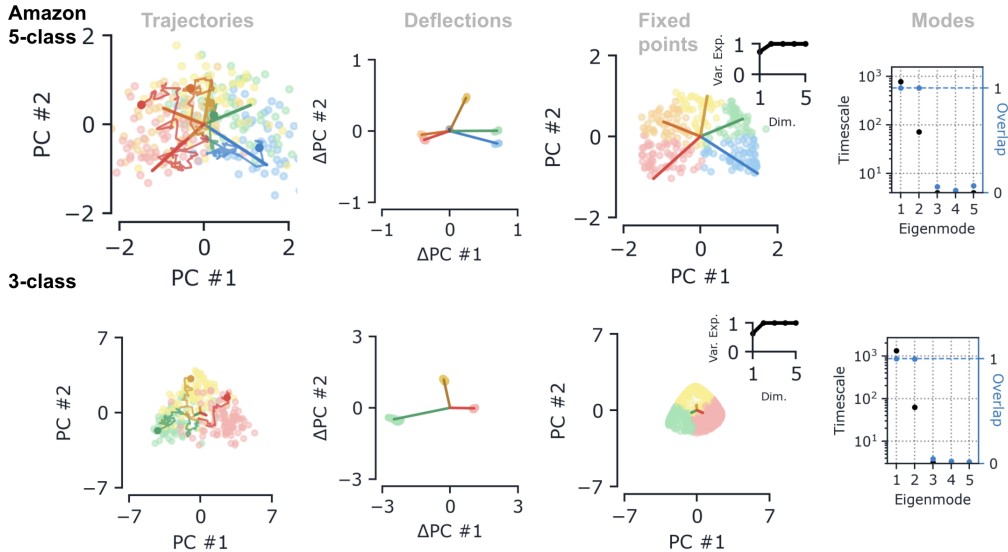

Figure 22: GRU on five-class and three-class Amazon

## F  THE EFFECT OF $\ell_2$ REGULARIZATION: COLLAPSE, CONTEXT, AND CORRELATIONS

Regularizing the parameters of the network during training can have a strong effect on the dimension of the resulting dynamics. We describe this effect first for the datasets with ordered labels, **Yelp** and **Amazon** reviews. We penalize the squared $\ell_2$-norm of the parameters, adding the term $\lambda ||\theta||_2^2$ to the cross-entropy prediction loss; $\lambda$ is the $\ell_2$ penalty and $\theta$ are the network parameters.

**Collapse**: Figure 25 shows the performance of the LSTM, GRU, and UGRNN as a function of the $\ell_2$ penalty. As the $\ell_2$ penalty is varied, the test accuracy usually decreases gradually; however, at a few values, the accuracy takes a large hit. The first two of these jumps correspond to a decrease in the dimension of the integration manifold from 2D and 1D and then 1D to 0D. The resulting 1D manifold is shown, for the example of a GRU on the Amazon dataset in Figures 26. The effects of collapse on the other architectures for the ordered datasets are identical.

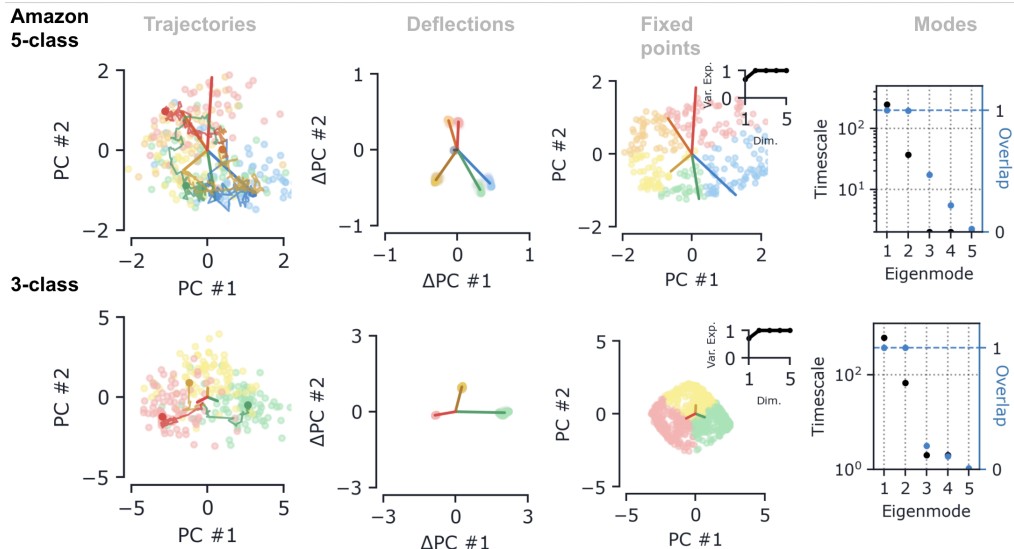

Figure 23: UGRNN on five-class and three-class Amazon

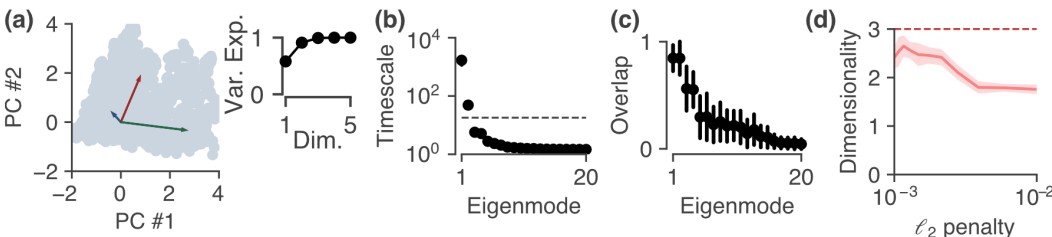

Figure 24: Analyzing networks trained on a 3-class version of the GoEmotions dataset. **(a)** Approximate fixed points (gray circles) and readout vectors. Inset shows the variance explained by the different principal components. The dynamics are (largely) 2D. **(b)** Across these fixed points, we see two slow time constants. **(c)** The eigenvectors aligned with the slow modes are aligned with the top PCA subspace. **(d)** Across networks trained with different amounts of regularization, we see the dimensionality (measured by participation ratio) is between 2 and 3, which reduces to less than two as the regularization penalty is increased.

When the regularization is sufficient to collapse the manifold to a 1D line, the dynamics are quite similar to the 1D line attractors studied in Maheswaranathan et al. (2019). A single accumulated valence score is tracked by the network as it moves along the line; this tracking occurs via a single eigenmode with a time constant comparable to the average document length, aligned with the fixed-point manifold. The difference between the binary- and 5-class line-attractor networks are largely in the way the final states are classified; in the 5-class case, the line attractor is divided into sections based largely on the angle the line makes with the readout vector of each class.

The collapse to a 0D manifold with a higher $\ell_2$ penalty is most strikingly seen in the recurrent Jacobian spectra at the fixed points (Figure 27). Here there are no modes which remember activity on the timescale of the mean document length. Given this lack of integration, it is unclear how these networks are achieving accuracies above random chance.

**Context**: While the focus of this study has been on how networks perform integration, it is clear from the plots in Figure 25 that the best-performing models are doing more than just bag-of-words style integration. When the order of words in the sentence is shuffled, these models take a hit in accuracy. Interestingly, when the $\ell_2$ coefficient is increased from the smallest values we use, the contextual effects are the first to be lost: the model's accuracy on shuffled and ordered examples becomes the same.

Understanding precisely how contextual processing is carried out by the network is an interesting direction for future work. It is important to show, however, that the basic two-dimensional integration

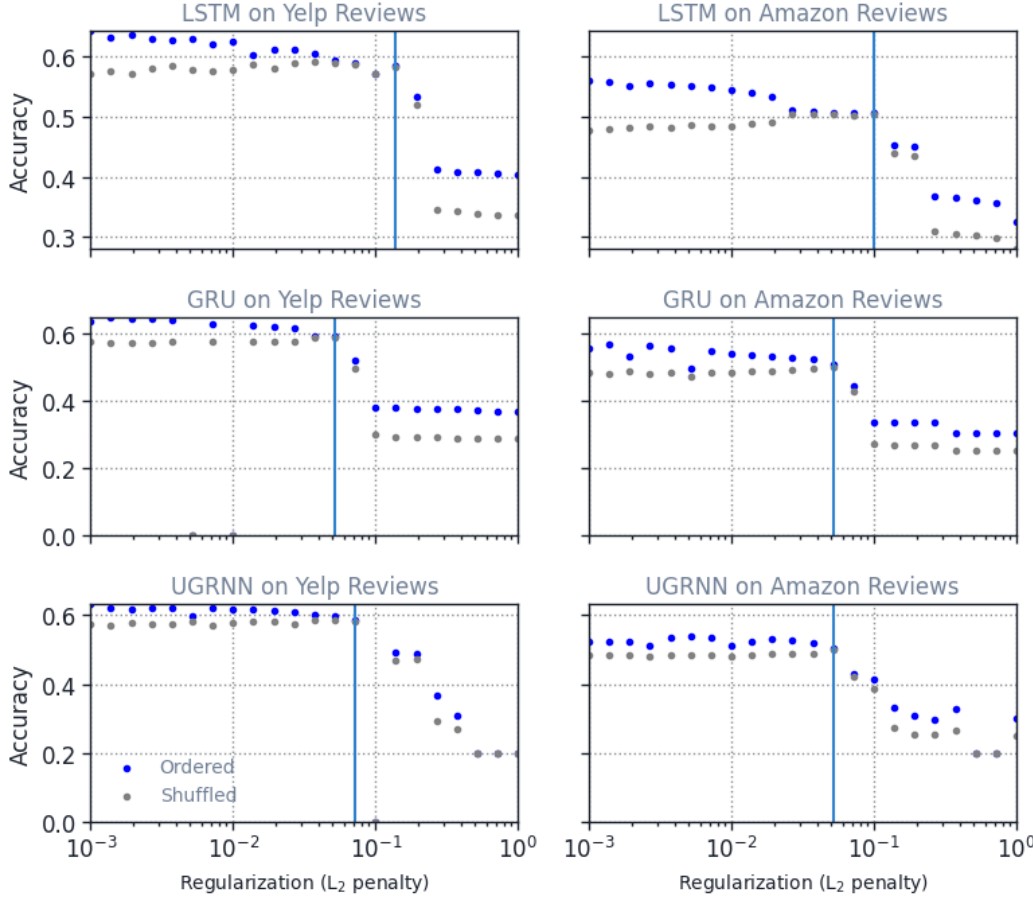

Figure 25: Performances of the LSTM, GRU, and UGRNN on ordered five-class datasets (both Yelp and Amazon reviews) as a function of $\ell_2$ regularization. Networks shown in the main text and appendix section E are highlighted with a line.

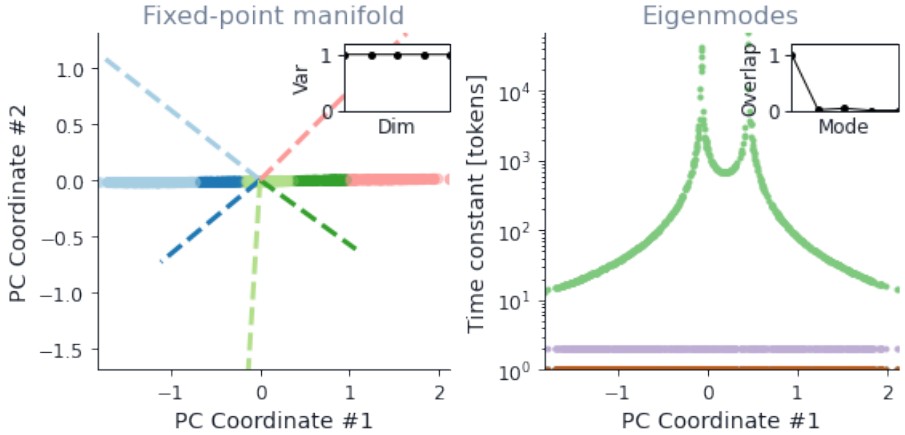

Figure 26: Geometry of a GRU trained on the five-class Amazon dataset, which due to high $\ell_2$ penalty $\lambda$ (here $\lambda = 0.72$) has collapsed to a 1D manifold, rather than the 2D manifolds seen in higher-performing models with lower $\ell_2$ penalty. This 1D collapse is seen in all models, in both the Yelp and Amazon datasets. Note the similarity of this fixed-point manifold to the line-attractor dynamics identified (in binary sentiment classification) in Maheswaranathan et al. (2019).

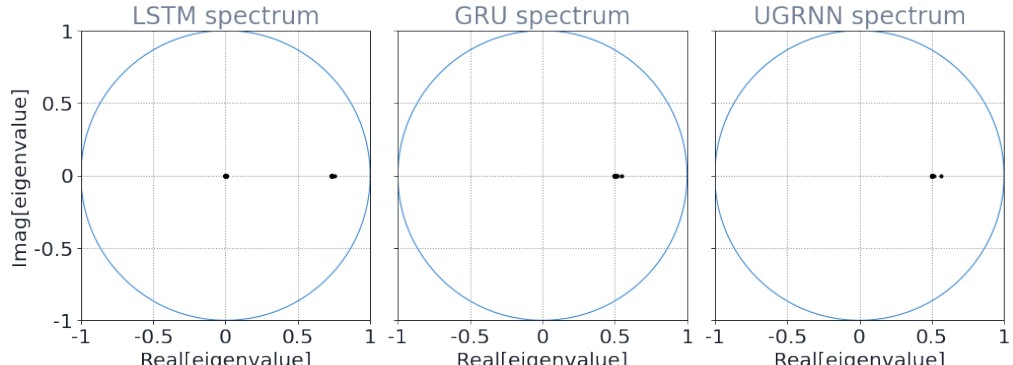

Figure 27: Jacobian spectra at an arbitrary fixed point of networks, trained on Yelp, with sufficiently high $\ell_2$-regularization penalty $\lambda$ to force the classification manifold to collapse to zero-dimensional (above, $\lambda_{\mathrm{LSTM}} = 0.268$, $\lambda_{\mathrm{GRU}} = 0.1$, $\lambda_{\mathrm{UGRNN}} = 0.268$). These spectra display no modes which can integrate information on the timescale of document length, i.e. no modes with close to unit magnitude.

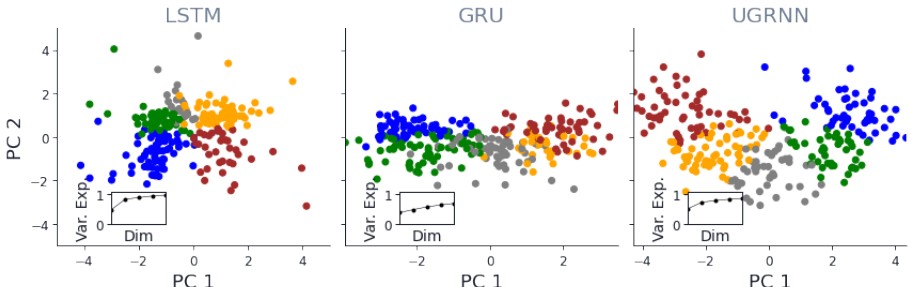

Figure 28: Fixed-point manifolds and predictions for RNNs trained on five-class Yelp with lower $\ell_2$ penalties than used in the main text. These networks outperform bag-of-words models and suffer a performance hit upon shuffling the test sentences, indicating an ability to process context (through a mechanism which we have not studied in this paper). The key point is: though these manifolds clearly extend into more than two dimensions (as indicated by the PCA explained-variance insets, the classes are nearly separable just by using the top two principal components. Thus, the mechanisms we identify in the main text still seem to underlie the operations of these networks, with contextual processing occurring on top of this integration.

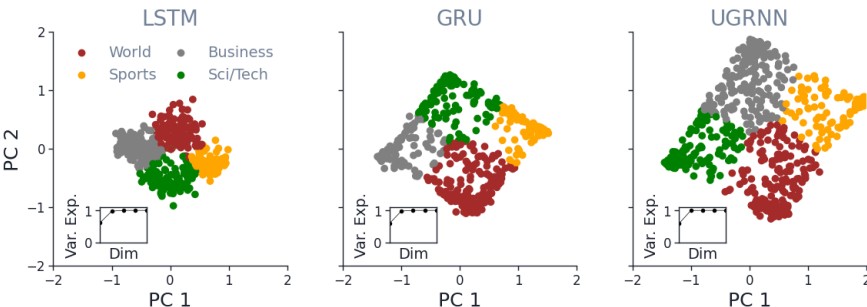

Figure 29: Fixed-point manifolds of LSTM, GRU, and UGRNN trained on 4-class AG News, with sufficiently high $\ell_2$ regularization penalty $\lambda$ to collapse the manifold from a tetrahedron to a square (above, $\lambda_{\text{LSTM}} = 0.3$, $\lambda_{\text{GRU}} = 0.1$, $\lambda_{\text{UGRNN}} = 0.3$). Across 10 random seeds per architecture, we observe the vertex of the fixed-point square corresponding to the Sports category always opposite to either the Business or Sci/Tech categories. This fact reflects correlations among the classes, shown in Figure 30. **Insets**: Variance explained by dimension after PCA projection, showing the 2D nature of the manifold.

mechanism we have presented in the main text still underlies the dynamics of the networks which are capable of handling context. To show this, we plot in Figure 28 the fixed point manifold, colored by the predicted class. As with the models which are not order-sensitive, the classification of the fixed points depends largely on their top two coordinates (after PCA projection). This is the case even though the PCA explained variance clearly shows extension of the dynamics into higher dimensions. It is thus likely that, similarly to how Maheswaranathan & Sussillo (2020) found that the contextual-processing mechanism was a perturbation on top of the integration dynamics for binary sentiment classification, the same is true for more finely-grained sentiment classification.

**Correlations**: As might be expected, increasing $\ell_2$ regularization also causes collapse in models trained on categorical classification tasks. For example, as shown in Figure 29, the tetrahedral manifold seen in 4-class AG News networks becomes a square at higher values of $\ell_2$, collapsing from three dimensions to two. That is, instead of class labels corresponding to vertices of a tetrahedron, when the $\ell_2$ regularization is increased, these labels correspond to the vertices of a square.

Interestingly, in the the collapse to a square, we find that — regardless of architecture and across 10 random seeds per architecture — the ordering of vertices around the square appears to reflect correlations between classes. Up to symmetries, the only possible ordering of vertices around the square are: (i) World $\rightarrow$ Sci/Tech $\rightarrow$ Business $\rightarrow$ Sports, (ii) World $\rightarrow$ Sci/Tech $\rightarrow$ Sports $\rightarrow$ Business, and (iii) World $\rightarrow$ Sports $\rightarrow$ Sci/Tech $\rightarrow$ Business. In practice, we observe that most of the time (26 out of 30 trials), order (iii) appears; otherwise, order (i) appears. We never observe order (ii).

To show how this ordering arises from correlations between class labels, we train a bag-of-words model on the full 4-class dataset. Taking the most common 5000 words in the vocabulary, we plot, in Figure 30, the changes in each logit due to these words. As the figure shows, for most pairs of classes there is a weak negative correlation between the evidence for the pair. However, between the classes "Sports" and "Business", there is a strong negative correlation ($R$=-0.81); between "Sports" and "Sci/Tech", there is a slightly weaker negative correlation ($R$=-0.61). Stated another way, words which constitute positive evidence for "Sports" are likely to constitute negative evidence for "Business" and/or "Sci/Tech". This matches with the geometries we observe in practice, where "Sports" and "Business" readouts are 'repelled' most often, and otherwise "Sports" and "Sci/Tech" are repelled.

# G  A CLOSER LOOK AT THE SLOW ZONE

In the main text, we were interested in *approximate fixed points*, or points which were slow on the timescale given roughly by the average length of documents $T_{\text{av}}$ in the training dataset. For practical purposes, points which are slow on this time scale can be treated as effectively fixed, since evolving the system for $T_{\text{av}}$ will result in little motion. Whether any of the points in the slow zone are exact

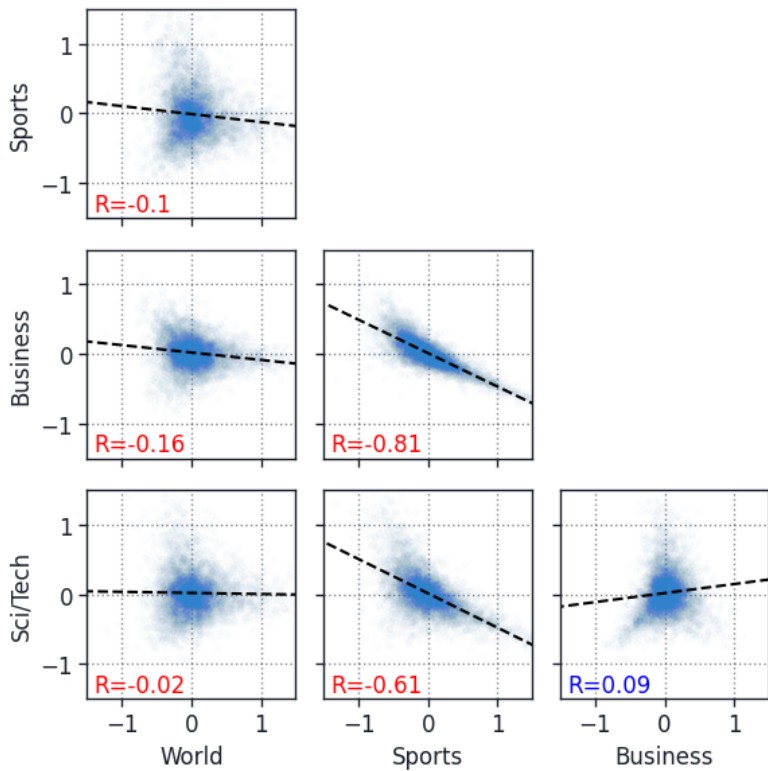

Figure 30: Correlations between evidence, provided by individual tokens, for the four classes present in the AG News dataset. The most significant correlations, seen in the middle column, are the negative correlations between the category pairs {Sports, Business} and {Sports, Sci/Tech}. These correlations influence the geometry of networks trained with sufficiently high $\ell_2$ penalty to collapse the manifold to a square (see Figure 29).

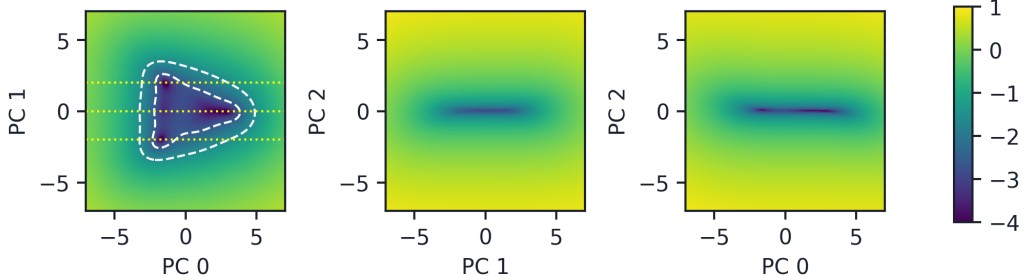

Figure 31: A view of the speed variation across the triangular manifold corresponding to a GRU trained on 3-class AG News (see Figure 1 in the main text). Colors are given by the log of the speed, see colorbar at the right. Points within the outer contour line correspond to speeds less than the inverse of the average document length; points within the inner contour are ten times slower still. Vertical slices in the middle and rightmost plot show the planar character of the manifold. The three horizontal dotted lines in the leftmost panels correspond to slices plotted in Figure 32.

.

fixed points of the dynamics, or whether all of them are simply slow points, is not something our numerical experiments are capable of resolving. However, we do find that there is structure to the slow zone in that all of the points are not uniformly slow.

We define the speed of a point, $S(\mathbf{h})$, as the distance traveled from that point after a single application of the dynamical map $F(\mathbf{h}, \mathbf{0})$ (Sussillo & Barak, 2013),

$$S(\mathbf{h}) = \|\mathbf{h} - F(\mathbf{h}, \mathbf{0})\|_2 \ . \tag{18}$$

Note that, as in the main text, we are characterizing the RNN with zero input (the autonomous system). In Figure 31, we plot three orthogonal slices of this function for a GRU network trained on the 3-class AG News dataset. All the points in the roughly triangular region we identified in the main text have speeds less than a tenth of the inverse document length, and are thus slow. However, there are a few neighborhoods, roughly located near the vertices of the triangle that are even slower. This structure can be seen further in Figure 32, in which we plot the speed as a function of the PC 0 coordinate for three values of the PC 1 coordinate. Similar structures can be seen in fixed point manifolds for other networks and datasets. Fully exploring these structures and understanding how they arise from the equations governing the RNN is an interesting direction for future research.

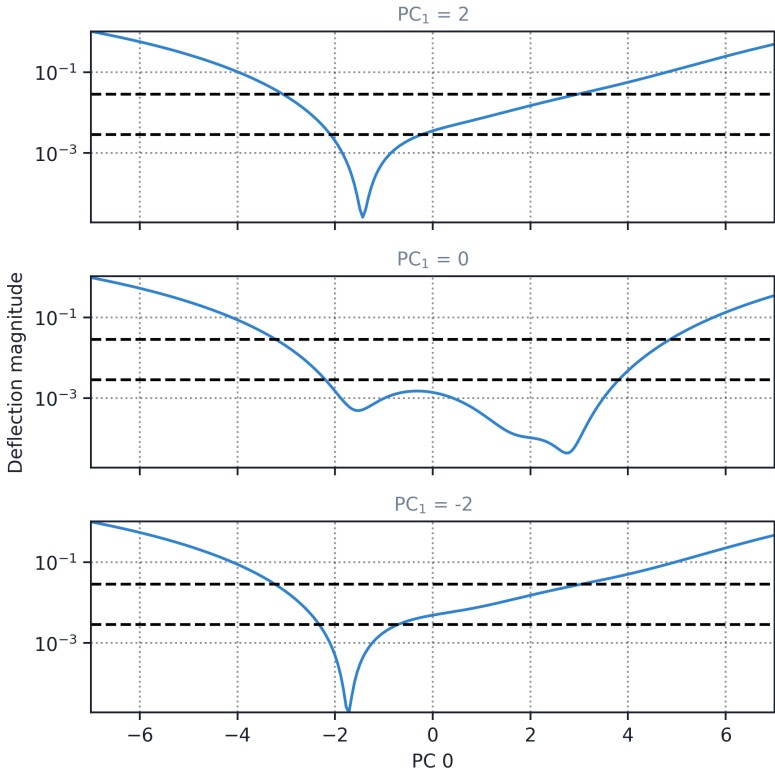

Figure 32: Speed versus PC 0 coordinate, plotted for various values of PC 1, corresponding to the three yellow lines in Figure 31. The dashed lines correspond to speed equal to the inverse document length, and one-tenth of that value.

.

