# OpenReview forum: "The geometry of integration in text classification RNNs"
_ICLR.cc/2021/Conference — ICLR 2021 Poster_

### Official Review · AnonReviewer3 · 2020-10-27
**Recommendation to reject**

**Rating:** 5
**Confidence:** 4

**Review:**

This paper presents an analysis on the trained recurrent neural networks (RNN) especially for NLP classification problems. The analysis takes the dynamical systems point of view and investigates the dynamics by looking at the Jacobians around the fixed points. This work founds low dimensionalility and attractor dynamics in the RNNs which might lead to a better undertanding of RNNs.

After reading the manuscript, I am leaning on not accepting it. My comments are below.

Opening the blackbox of RNNs is an important topic in machine learning and neuroscience. This work uses the tools from dynamical systems and focus on the RNNs in NLP classification problems. It empirically shows the connection between the dimensionality of internal dynamics and the tasks.

The weakness of this work is the lack of theoretical implications. This paper put a lot of work on showing low-D dynamics are embedded in the high-D RNN for low-D tasks that is already shown in many empirical studies. The question is if one can train a low-D RNN to perform the same task, and why or why not.

On the integration of evidence, the possible mechanisms include, for example, fixed points, line attractors or sequences of stable fixed points (forming a path to the decisions). Further investigation is needed to discriminate between them beyond only analyzing the Jacobian at the final states.

Minor comments
- sec 3.1 do the synthetic data contain inconclusive phrases (all words are 'neutral')?

- Fig 2. The subfigures are wrong-labeled.

- A typo in 'As the position withing...' in page 5

- arxiv:1906.01005 on continuous-time GRU might be relevant.

---

> ### Author Response · Authors · 2020-11-13
> **Reply to Reviewer 3**
>
> We thank the reviewer for their critical assessment of our work.
>
> 1.  “This paper put a lot of work on showing low-D dynamics are embedded in the high-D RNN for low-D tasks that is already shown in many empirical studies.”
> While it is true that there are a number of tasks where trained high-dimensional continuous-time RNNs exhibit low-dimensional dynamics, we are unaware of work that demonstrates this for modern, discrete-time RNNs, beyond the previously-cited works of Maheswaranathan et al (2019) and Maheswaranathan and Sussillo (2020).  Our thorough experiments confirm this phenomena on a wide variety of different classification tasks in NLP, extending the previous binary sentiment analyses.  Moreover, we demonstrate that the geometry of the integration manifold has a particular structure, and show how this structure depends on task structure. We believe this understanding constitutes a significant advance both in terms of our understanding of NLP classification tasks, as well as in understanding of how low-D RNN dynamics vary depending on properties of the classification task or dataset. These are fundamental scientific findings that constitute the main contributions of our paper.
> If other works studying similar phenomena in discrete-time RNNs exist in the literature, we would be happy to cite them in our paper.
> 2. “The question is if one can train a low-D RNN to perform the same task, and why or why not.”
> The question of whether or not one can train a low-D RNN directly is related, but tangential, to the scientific goals of our investigation. Our findings do strongly suggest that one can distill a low-D RNN that retains nearly the same accuracy. In fact, our analysis predicts the exact dimensionality needed to perform such a distillation.
> 3. "On the integration of evidence, the possible mechanisms include, for example, fixed points, line attractors or sequences of stable fixed points (forming a path to the decisions). Further investigation is needed to discriminate between them beyond only analyzing the Jacobian at the final states”
> Thanks for this feedback.  As a technical point, our analysis relies on numerically finding approximate fixed points for the entire network---we are not just analyzing the Jacobian at the final states.
> We have added plots to Appendix G in our revision which show the slowness of certain fixed point manifolds persist for entire planar slices, which provides direct evidence that the integration happens along approximate plane attractors, *not* a series of distinct fixed points. By “approximate” plane attractor, we mean that the dynamics along the attractor are slow relative to the timescale of the task. This means that any point along the attractors is effectively fixed over the number of timesteps that the network is required to operate over.  Thus far, we have not found evidence contradicting the presence of a continuous slow manifold.
> 4. “sec 3.1 do the synthetic data contain inconclusive phrases (all words are 'neutral')?”
> Yes, the synthetic data does contain inconclusive phrases, including phrases where all words are neutral or there are multiple occurrences of a maximum score. In these cases, the tie is broken by assigning the score to the class with the lowest index. We have added a footnote to Appendix D.1 to clarify this point.
> 5. “arxiv:1906.01005 on continuous-time GRU might be relevant.”
> Thank you for bringing this work to our attention; it is indeed relevant, and we have cited it in our updated related work section.

---

> > ### Comment · AnonReviewer3 · 2020-11-17
> > **Thanks for reply**
> >
> > Thanks for addressing my comments.
> >
> > Cueva et al, PNAS, 2020 talked about low-D dynamics in RNNs for low-D tasks. Though the tasks were not NLP classification, it might be relevant.

---

### Official Review · AnonReviewer2 · 2020-10-28
**An important step towards interpreting RNNs trained on NLP tasks**

**Rating:** 8
**Confidence:** 4

**Review:**

This paper examines RNNs trained to perform a range of text classification tasks, and demonstrates that they can be understood using a dynamical-systems analysis. Specifically, the paper shows that RNN dynamics explore low-dimensional subspaces determined by the categories to be distinguished. Within those low-dimensional subspaces, RNN dynamics integrate evidence towards each category. The approach is based on recent works by Maheswaranathan et al, but extends them to a large range of tasks. This is a big step towards interpreting RNNs trained on NLP tasks, and I strongly recommend it for ICLR 2021.

Strengths:
- a strong conceptual framework for analysing trained RNNs
- large battery of categorisation tasks on both natural and synthetic datasets
- clear and compelling interpretation of trained RNNs.


Concerns:
The results for ordered classification (Fig 4) are a little puzzling. One would expect that 1-star and 5-star reviews should be further away from each other in PC space than 1-star and 3-star reviews. Related to this, I don't understand why "sentiment" and "intensity" scores should be orthogonal - intuitively they do not seem to be orthogonal for the words used in the synthetic data. One possibility is that the specific 2d organisation in Fig 4 results from the manner in which the readouts were implemented. Presumably, the readouts were implemented as 5 independent categories as in other tasks? If that's the case, the readout structure does not necessarily take into account the ordering in the task. A different readout structure (eg 5 different levels on a single readout) might lead to a different low-d structure.

Other feedback:
- I did not quite understand what is plotted in Fig 4c and similar. The legend says "fixed points", but are these fixed-points in response to different inputs? Which inputs? What do coloured regions and coloured lines respectively represent?
- how important is non-linearity in these tasks? Eg how big is the difference in performance between RNNs and linear accumulation models based on LSA?

Related work:
- a recent paper by Schuessler et al (arXiv:2006.11036) suggests that the low-d dynamics in the sentiment-classification task can be traced to low-rank structure in connectivity, as in neuroscience tasks.

---

> ### Author Response · Authors · 2020-11-13
> **Reply to Reviewer 2**
>
> We thank the reviewer for the positive feedback on our work.  Below, we respond to each point raised by the reviewer.
>
> 1. Regarding the puzzling results for ordered classification:
> The 1 and 5-star reviews being further away is indeed what we expected going into this task. In fact, we expected an extended line attractor, similar to what was seen for sentiment analysis in Maheswaranathan et  al., (2019) and Maheswaranathan & Sussillo (2020), with the 1 and 5 star reviews being at the extreme ends of this line attractor. Our interpretation of the fact that the network’s solution appears as a 2-dimensional integrator is that the easiest way for the network to classify this particular task is for it to keep track of intensity in addition to sentiment. If this really is something that is useful for the network to classify this data, we took this to mean our preconceived notion of 1 and 5 star reviews being further from one another than 1 and 3 star reviews might not be correct. That is, it appears the network learns that 1 and 5 reviews are fairly close to one another in intensity even though they are further away in sentiment. Meanwhile, it learns a 1 and 3 star review are probably closer in sentiment, but further away in intensity.
> The way we like to think about “sentiment” and “intensity” axes is that each word can have some degree of both of these features, so said word can move you in any direction in the two-dimensional space they form.  For example, in the synthetic dataset, by design we made it so the words “bad” and “awful” both convey negative sentiment, but the former contains less intensity than the latter. So although the axes along which the network keeps track of “sentiment” and “intensity” may be different, these words can move you along both axes simultaneously. We see this in the natural dataset as well, where for example, the word “horrible” moves you toward negative sentiment and greater intensity.
> We did implement the readouts as five independent categories, and this is agnostic to the inherent ordering of the dataset. We felt that using the same readout structure as what we would use in, say, a 5-class categorical dataset, would allow us to better compare how the dataset affects the underlying manifold while holding network architecture constant. It would certainly be interesting to see how a readout structure that inherently takes into account the ordering affects the hidden state space.
>
> 2. “a recent paper by Schuessler et al (arXiv:2006.11036) suggests that the low-d dynamics in the sentiment-classification task can be traced to low-rank structure in connectivity, as in neuroscience tasks.”
> Thank you, we were not aware of this work.  We have now mentioned it in our related work section, and have removed our statement suggesting that the previous works we cited were the only ones to our knowledge which studied discrete-time RNNs.
>
> 3. “How important is non-linearity in these tasks? Eg how big is the difference in performance between RNNs and linear accumulation models based on LSA?”
> On these tasks, linear accumulation or bag-of-words models can do quite well.  Depending on the task, we see a small performance gap between RNNs and linear accumulation models (~2-4%).   This is in line with previous work for binary sentiment classification (Maheswaranathan & Sussillo 2020). We do suspect this additional improvement by the nonlinear models can be understood independently of the integration or accumulation mechanism (as demonstrated in the binary case in Maheswaranathan & Sussillo 2020). The focus of our work was on the geometry of the linear integration manifold, which is responsible for nearly all (over 90%) of the total accuracy.
>
> 4. “I did not quite understand what is plotted in Fig 4c and similar. The legend says "fixed points", but are these fixed-points in response to different inputs? Which inputs? What do coloured regions and coloured lines respectively represent?”
> Thanks for pointing out that this was unclear.  We have added text to the captions of Figures 1, 4, and 5 to address these issues. In Figure 4c and similar, we plot fixed points of the system in response to zero input.  Fixed points of the system corresponding to the average input are similar.  The color of each point corresponds to the label which the network assigns that point, to show that the network uses the position of the point within the 2D plane to perform its classification.  The lines, as in 4a and 4d, are the corresponding readout vectors for each class.

---

### Official Review · AnonReviewer4 · 2020-10-28

**Rating:** 7
**Confidence:** 5

**Review:**

The submission furthers existing dynamical systems inspired analyses of recurrent neural networks with focus on text classification.
The motivation is well founded, and stems from the wide-spread applicability and success of RNNs in NLP tasks. The efficacy of this class of models, is nevertheless lacking a mechanistic understanding of the properties that have enabled their success.

The major contribution of this submission is an in-depth analysis of the types of dynamics that permit the **trained** networks to solve text classification tasks. The conclusions notably include:

1. Empirical evidence that the networks use low-dimensional attractors ($ N\ll d$ where n is the number of, potentially not mutually exclusive, classes and d is the hidden layer size). This is result is of twofold importance, in as much as it reveals a fundamental mechanism that models leverage but also because it could be used as motivation to study the compression of large recurrent models to lower-dimensional and hence more interpretable and more energy efficient equivalents.

1.  The geometry of the hidden-state representations is dictated by the task, and loss function. To the best of my knowledge, such a result hasn't been stated as clearly. It bears similarity to results from theoretical neuroscience, where the embedding dimensionality of dynamical systems has been shown to be upper-bound by the 'complexity' of the task (see Gao et al 2017).

Despite the strong, and convincing empirical contributions, the manuscript suffers from some shortcomings. I would like to emphasize at this point, that I do not believe they invalidate the empirical results and hence the essential contribution. That being said, I find the following most concerning:

The analysis is contingent on numerical root finding to locate the fixed points of a *inhomogeneous* map. The problem certainly presents a substantial theoretical challenge, but the specifics of the analysis require further motivation or a modification of the methods.
1. Specifically, the analysis of the inhomogeneous system is performed by looking at the linearization of the dynamics around the terminal time point. However, the stability of the inhomogeneous system generally depends on the entire sample path of the input process $x_t\, \forall t \in [0,\ldots T]$.
1. Moreover, the authors further simplify the analysis by assuming that there exists a prototypical terminal $x_T$ which is given by the average input. This strikes me as a peculiar choice, given that the maps are potentially highly non-linear and averaging ought not to commute with the application of the map. I personally would prefer a Monte Carlo estimate of the `hidden deflections` using samples from the datasets.
1. On a technical level, the linearization analysis depends on conditions analogous to those stipulated by the Hartman-Grobman theorem from differential equations. I.E. the stability of the system is only given if no generalized eigenvalues are purely imaginary. While it may seem like a pedantic distinction, it bears relevance upon the conclusion regarding the existence of ` plane attractors`.  If the integration is indeed performed by a linear submanifold that has zero 'flow', then it's stability cannot be analyzed using the Hartman-Grobman type approach.
1. The previous difficulty is compounded by the fact that all the equilibria and the plane attractor are only inferred with finite numerical precision, which muddles the distinction between a 'true' integrator plane attractor with zero motion within it, and a slow linear submanifold. This problem could potentially be ameliorated by computing the linearizations with arbitrary floating point precision, rather than the 16/32/64 bit precision that are default in neural network libraries.

The following are minor comments:
1. Building on remark (4), it is entirely not clear to me whether the detected submanifold is indeed a center manifold of a single attractor or rather a union of topologically disconnected equilibria. This distinction is arguably inconsequential from a practical point of view, but additional experiments investigating this would benefit the theoretical understanding of the problem.
1. In Fig. 1 the initial points are only marked for the artificial data. Moreover, for the real data the trajectories seem to start from different locations.
1. In Fig. 2 the right-hand side y-axis is not explained clearly, neither in the main text nor in the caption.
1. The model specification for GRU's and UGRNN's leaves out what specific non-linearity was used.
1. The figures in the main text lack indication of what specific model generated the data. I assume this was done to emphasize the universality of the dynamics, but I find it makes it harder to reproduce the result should it be necessary.
1. The network hidden unit sizes were chosen to be different for different architectures -- it would be good to state why this was done, should the result depend on the hidden unit dimensionality.
1. The relation between task/loss and the arrangement of terminal hidden states is very striking, and I think the explanation in its current form is sufficient. At the same time, I'm curious to what extent does the loss necessitate the structure of the manifold. For instance, for dynamics with a non-compact phase space (e.g. linear systems) does the use of cross-entropy somehow manage to contain the hidden states to simplices? If not, to what extent does this depend on the choice of non-linearity for the RNN units?

##### Overall, I think the submission is an interesting contribution, with well thought out empirical experiments. Moreover, it is well written and clear. Pending the authors response, I am rating it as marginally below the threshold.

---

> ### Author Response · Authors · 2020-11-13
> **Reply to Reviewer 4**
>
> We thank the reviewer for the thorough feedback, and very much appreciate the detailed engagement with our work.  We have performed additional measurements in response to the feedback and added a section to our Appendix (details below).
>
> Before we get into point-by-point responses, we would like to clarify something which is germane to multiple points raised by the reviewer.  When we describe dynamical structures such as fixed points and plane attractors in the text, we mean that they are fixed to a level of approximation sufficient for the task the network is trained on.  In these NLP classification tasks, for example, there is a natural time scale given by the maximum length (in tokens) of a document in the training set.  We take approximate fixed points as points where the drift (zeroth order term) of the dynamics are slow on this time scale.  These points are not exact fixed points of the dynamical map, and we have added language to our paper to clarify that we are not claiming this; we thank the reviewer for making us be more precise in our language.  Similarly, when we refer to a plane attractor, we mean an approximate plane attractor, where the network drift is small over the entire plane (again, relative to the task’s time scale).  For our practical purpose of understanding how the network solves the classification task, this level of approximation is sufficient.  A more precise characterization of the dynamics of trained RNNs is an interesting question in its own right, but not quite the central motivation of this paper (although one interesting line of work along this thread is arxiv:1906.01005, pointed out by another reviewer).
>
> Similarly, regarding the eigenvalues of the linearized system, the integrator modes we find have eigenvalues which are approximately unity, but as with the drift, this approximation is valid to the timescale of the task: the time constant of the integrator eigenmodes we find is large relative to this timescale, but not infinite, since the network during training has no pressure to make these modes much longer-lived than the task demands.
>
> We have modified the text to elaborate on the fact that we are talking about approximate fixed points and approximate plane attractors when we introduce these concepts in the manuscript, along with corresponding discussion of why the approximate structures are still useful to study.  While our analysis only relies on these approximations, we believe this distinction between approximate and exact dynamical structures is important, and appreciate the reviewer raising the concern.
>
> Reply to major comments:
> 1. To clarify, the linearization is performed about approximate fixed points, not the terminal time points of the trajectories. These fixed points are found numerically by minimizing the speed of a point, starting from all hidden states the network visits while processing a given test batch. In practice, we find the hidden state space the network occupies during the entire sample path of the input process to be close to the fixed point manifold. Another reviewer also seemed to have this misconception, so we have added additional text around the discussion of Figure 2 as well as in the captions of Figures 1, 4, and 5 to more explicitly clarify what is being plotted.

---

> > ### Author Response · Authors · 2020-11-13
> > **Reply to Reviewer 4, continued**
> >
> > 2. This is a very important point, and we should clarify our choice to study the behavior of the autonomous dynamical system (i.e. the RNN update equation with the zero-vector as input).  First, regarding language: we realize that our footnote mentioning that the average input was well approximated by the zero-vector might be taken to mean that we were attempting to get an understanding of the “average” system dynamics by studying the dynamics with the average input.  This is not the case; as the referee points out, the nonlinear map will not commute with averaging.  We have removed that language from our footnote, clarifying that we are choosing to study the particular dynamical map corresponding to zero input.
> > It is true in general that the autonomous dynamical system need not be representative of the system’s behavior in the presence of inputs (x_0, … x_t), especially for highly nonlinear dynamical systems. However, we can verify that the approximate attractors identified by our analysis in the absence of input is a good description of the system’s behavior, even in the presence of inputs. We can do this in a couple of ways. First, by training purely linear bag-of-words models, we find that a purely linear integration mechanism achieves good performance on these tasks (around 90% of the nonlinear RNN’s accuracy), suggesting that the linear integration mechanism is sufficient to describe a large fraction of the system’s behavior. Second, we computed the relative error of the linear approximation to the full nonlinear system dynamics (using a Jacobian that is computed in response to no input) and found that it is quite small over a large random sample of possible inputs (to within a few percent). Thus, we can make post-hoc justifications for studying system behavior in response to zero input.
> >
> > 3. Let us restate this point to make sure we have understand the concern correctly.  In continuous-time systems, the Hartman-Grobman theorem states that if a fixed point has an eigenvalue which is purely imaginary, then the stability of the fixed point cannot be determined purely by analyzing the linearized dynamics.  By analogy, in discrete-time systems if a fixed point has an eigenvalue which has exactly unit magnitude, the stability cannot be determined by the linearized dynamics.
> > If we have indeed understood the worry correctly, then we think it is not an issue in our case, for a reason similar to that described above: none of the eigenvalues we find have exactly unit magnitude.  Their time constants, while long relative to document length, are finite.  This deviation from unit magnitude is well beyond numerical precision.  In this case, a linear stability analysis, such as the one we have conducted, should be sufficient.
> >
> > 4. We agree that finite numerical precision is in principle a difficulty, but as mentioned in the response to the previous point, the deviations of the eigenvalues we find from unity go well beyond numerical error; thus we do not see true plane integrators with zero motion, but indeed slow manifolds.  Given the lack of pressure on the network to learn true integrators, the existence of these slow manifolds (slow, again, relative to the task’s timescale) is in line with expectations: as far as the loss can tell, there is not much of a difference between a perfect integrator and a slow manifold.
> >
> > Minor comments:
> > 1. Thanks for pointing this out.  While the focus of our paper was on a practical ‘working understanding’ of the RNN dynamics, we are happy to provide additional experiments which may benefit a theoretical understanding of the problem.  We have added plots to Appendix (section G) showing the speed of points across the slices of the fixed point manifold.  These plots show that there is a definite variation of speed across the approximate fixed-point plane, though all points within this plane are very slow on the time scale of the document length.  If there are other measurements we could perform which would help achieve a more thorough understanding of these systems, we would be happy to add them.
> > 6. Thanks for pointing out that this was unclear.  Since the LSTM has two parts to the hidden state, a memory and a ‘state’ (for lack of a better word), we thought for consistency it would be best to keep the dimension of the hidden state fixed across all architectures, and use the same dimension for the memory.  We have updated Appendix B to describe this more precisely, though we do not believe that the precise value of hidden unit dimensionality should make a large difference.
> > 7. This is an interesting question and we would very much like to continue to explore dependencies upon loss, non-linearities, task, and architecture in future work. There is some discussion in Appendix D.1 speculating on how the presence of the soft-max in cross entropy produces the simplex shapes, but we have not yet performed any experiments with purely linear systems.
> >
> > 2,3,4,5: changes made in text

---

### Official Review · AnonReviewer1 · 2020-10-29
**Exciting work, but relation to earlier work should be presented more clearly. Weak reject, maybe accept with revisions.**

**Rating:** 7
**Confidence:** 4

**Review:**

### Summary:

Weak reject. This paper might be a weak accept if revised to present its key contributions more clearly.


### Reasons for score:

This paper is an interesting extension of the work from “Reverse engineering recurrent networks for sentiment classification reveals line attractor dynamics” (Maheswaranathan et al, 2019, https://arxiv.org/abs/1906.10720). The work presents new insights into the dimensionality and workings of low-dimensional attractors in which RNNs accumulate “evidence” in their hidden layer. The authors extend the above-mentioned work by Maheswaranathan et al from sentiment classification to other classification settings, such as multi-label classification. The authors find intriguing differences in the topology of these attractors for different types of classification. This work will be of interest to many researchers in the ML community.

My main gripe with the paper is that in many places, it does not clearly distinguish between existing work and the paper’s contributions. In my mind, the paper is an extension of work by Maheswaranathan et al (2019) and Maheswaranathan & Sussillo (2020). While the authors of the paper reviewed here cite both of these works, overall the paper reviewed here could be more clear about what its contributions are, and how they relate to earlier work. In particular, the abstract describes contributions from Maheswaranathan’s 2019 paper, but doesn’t mention that this is existing work.



### Pros

- The paper presents work on one of the most exciting topics being researched in this community today: how should we think about how neural networks (RNNs, in this paper) process information. The paper presents novel and exciting insights on this topic, and is one of a small number of contributions on the topic not just at this conference, but in general. The paper was a great read.
- The precise research question addressed (how do low-dimensional attractors in hidden layers of RNNs differ for different classification tasks) is a useful problem to investigate, and the author’s findings may inspire further research and insights into the mechanics of how RNNs make predictions.


### Cons

The authors should be more clear about the paper’s key contributions, which are currently somewhat buried in material that has already been presented in the above-mentioned Maheswaranathan  papers. Being clear about a paper’s contributions and how they relate to existing work makes the paper more helpful for readers. I believe that the authors meant no harm here, and they did cite Maheswaranathan’s existing work appropriately. As a reader, I ask myself “what are the key claims that the papers makes, and how does the paper support these claims?” These questions are answered in the text, but the distinction between existing work and new contributions should be clearer and mentioned earlier.

For instance, the first half of the abstract, up to and including "*Specifically, across architectures and datasets, RNNs accumulate evidence for each class as they process the text, using a low-dimensional attractor manifold as the underlying mechanism.*", could easily have been the abstract of Maheswaranathan’s 2019 paper. In this way, the abstract advertises material without pointing out that this material is in fact not novel. The last sentence of the abstract (“ *To the degree that integration of evidence towards a decision is a common computational primitive, this work lays the foundation for using dynamical systems techniques to study the inner workings of RNNs.* ”) also looks somewhat as if credit is claimed for work from Meheswaranathan’s 2019 paper, where they write ‘ *Input words with positive valence (“amazing”, “great”, etc.) incremented the hidden state towards a positive sentiment prediction, while words with negative valence (“bad”, “horrible”, etc.) pushed the hidden state in the opposite direction.* ‘.


### Questions during rebuttal period:

Please address be more clear about the paper’s key contributions, and how they relate to existing work (see “Cons” section and “Reasons for score” section above).


### Some suggestions:

I would recommend to include a statement like “we expand on earlier work \cite{Maheswaranathan 2019, Maheswaranathan 2020} on the  …. for sentiment analysis, and find that … “ right in the abstract.

To improve the readability of the paper, and better highlight the paper’s contributions, I would moreover suggest to rewrite “We find the network dynamics in these text classification…” paragraph and instead use an “Our main contributions are <itemized list follows>” format here. This way the reader of a paper can quickly glean the paper’s key contributions/claims, but also more easily form an opinion about how well the paper’s arguments support the key claims.

Small typos
- “withing: should be “within”
- “eigenvalues well” should be “eigenvalues are well”

---

> ### Author Response · Authors · 2020-11-13
> **Reply to Reviewer 1**
>
> We thank the reviewer both for the positive assessment of our work and the critical feedback on the writing.  We have implemented all of the reviewer’s suggestions in the attached revision, in order to make it apparent to the reader how our contributions differ from those of the mentioned previous works.
>
> Specifically, we have rewritten the paragraph beginning “We find the network dynamics…”, converting it instead into an itemized list of contributions as recommended.
>
> Second, we have added phrasing to the abstract to clarify that the dynamical systems perspective we use in this work follows from Maheswaranathan et al., (2019), including an explicit citation of that work. We further modified the final sentence of the abstract to make it clear that this work continues to lay the foundation for dynamical systems analysis, rather than being the first to lay the foundation.
>
> Finally, we have modified the paragraph directly above the “Our contributions” list to describe clearly the focus of this work on understanding the geometry and dimensionality of the integration manifold in various tasks beyond binary classification.  Our phrasing explicitly draws the line between our work and the recent works of Maheswaranathan et al., (2019) and Maheswaranathan & Sussillo (2020).  We believe the itemized list suggested by the reviewer also aids in this respect, since the contributions of this work are now under their own heading.
>
> We hope this sufficiently addresses the concern from the review. As this was the only concern, we hope that you will
> consider increasing your score to reflect the amended manuscript.

---

> > ### Comment · AnonReviewer1 · 2020-11-24
> > **Better paper, accept**
> >
> > ### Update during review period
> >
> > - With the revisions made by the authors, this is a much better paper now. I updated my review score to reflect this.

---

### Official Review · AnonReviewer5 · 2020-11-06
**Interesting paper with useful analyses**

**Rating:** 7
**Confidence:** 4

**Review:**

### Paper Summary

This paper sheds light on how trained RNNs solve text classification problems by analyzing them from a dynamical systems perspective. It extends recent work where a similar analysis was applied to the simpler setting of binary sentiment classification. When projecting the RNN hidden states to principal dimensions that explain most of the variance, the authors find (N-1) dimensional simplex attractors for N-dimensional classification, 2D attractors for ordered classification, and N-dimensional hypercubes for multi-label classification.

### Strengths

In general, the paper is clear and enjoyable to read.

The use of synthetic data strengthens the analysis by first showing what behavior one might expect from RNNs on "clean" data, and then comparing to what is observed on natural data.

The use of different categories of tasks not only makes the experimentation thorough, it also clarifies understanding by contrasting behavior across these categories.

Based on the presented dynamical systems analysis, I think practitioners can acquire valuable intuitions about how RNNs process information (at least for certain problems) that is "correct", i.e. it has been verified by sufficient experimentation.

Finally, I agree with the authors that the paper motivates further work into understanding the behavior of RNNs from this perspective.

### Weaknesses

I found the multi-label classification experiments on natural data to be limited, since the authors only used reduced 2- and 3-class versions of the GoEmotions dataset and the behavior was not as "clean" as that for synthetic data for the 3-class setting (App. E.5). This part of the analysis leaves me somewhat uncertain about general behavior to expect for multi-label classification in practice.

### Review Summary

I found this paper to be interesting, well-written and useful. I recommend an Accept, but the authors can improve the paper by strengthening the multi-label classification experiments.

### Post Author Response

I thank the authors for their hard work and I'm happy to see that the analysis has improved.

---

> ### Author Response · Authors · 2020-11-13
> **Reply to Reviewer 5**
>
> We thank the reviewer for the positive assessment of our work, and their suggestion to strengthen our understanding of the multi-label classification task.  Following this suggestion, we have improved our understanding and updated the relevant section in the paper.
> Specifically, we suspected the disagreement between the GoEmotions dataset and the synthetic data was due to class imbalance issues. GoEmotions, unlike the other datasets we studied, does not have nearly the same number of classes per label. For the synthetic data, however, the four possible label combinations in the 2-class case (or eight in the 3-class case) are perfectly balanced.
> We explored whether changing the class balance of the GoEmotions dataset would alleviate these differences. Indeed, it does. We have updated the natural dataset plots of Figure 5 for a balanced version of the GoEmotions dataset. We generated a balanced version by upsampling classes with small numbers of examples. This results in a better one-to-one comparison with the synthetic data and we see a significantly clearer match in their integration manifold.

---

### Author Response · Authors · 2020-11-13
**Changes to the revised draft**

Amongst many minor edits correcting typos and clarifications, edits include:

1. We have added an “Our Contributions” section to the introduction that more clearly states the results of this work. We have also updated the introduction to better distinguish the results of our work from previous works.
2. We have added several references suggested by reviewers that are of relevance.
3. Per a few reviewer’s suggestions, we have updated all figure captions in the main text to more clearly explain what is being plotted in each figure. We have also updated what is being plotted in a few figures to better stylistically match the synthetic and natural datasets.
4. We have updated Figure 5 with results that show a better match between synthetic and the GoEmotions dataset for the (2-label) multi-labeled data. Previously, the mismatch between the natural and synthetic data was due to the former having a large class imbalance (this is because there are significantly more phrases in the GoEmotions dataset that lack the two emotions we were probing). After balancing the GoEmotions dataset, we get the results shown in Figure 5. This change is in line with the matching achieved in Figures 1 and 4, where the synthetic and natural datasets were both equally balanced.
5. We have added Section G to the appendix showing the speed of points across the slices of the fixed point manifold.  Plots in this section show that there is a definite variation of speed across the approximate fixed-point plane, though all points within this plane are very slow on the time scale of the document length.

---

### Decision · Program_Chairs · 2021-01-07
**Final Decision**

**Decision:**

Accept (Poster)

**Comment:**

this paper adds onto the line of research in investigating the mechanism by which a recurrent network solves a supervised sequence classification problem, following the recent studies such as Maheswaranathan et al., 2019 and Maheswaranathan & Sussillo (2020). in doing so, this paper hypothesizes and confirms that the internal hidden state of a recurrent net, be it GRU or LSTM, evolves over a planar (approximate) attractor as it reads the input, amounting to integrating the evidence as it processes the input sequence, and demonstrates the existences of these attractors and integration dynamics on three types of problems (classification, ordered classification and multi-label classification.)

there were some potentially misleading or confusing statements throughout the manuscript in the initial version, which were pointed out by the reviewers. the authors however did a commendable job of addressing these concerns by the reviewers to the point that most of them have revised their scores up.

based on the reviewers' assessments, authors' response and their exchange, i strongly believe this work will enrich our understanding of recurrent nets further.